# Epigenetic regulation of the circadian gene *Per1* contributes to age-related changes in hippocampal memory

Janine L. Kwapis[1,2,3], Yasaman Alaghband[1,2], Enikö A. Kramár[1,2], Alberto J. López[1,2], Annie Vogel Ciernia[4], André O. White[5], Guanhua Shu[1,2], Diane Rhee[1,2], Christina M. Michael[1,2], Emilie Montellier[6], Yu Liu[7,8], Christophe N. Magnan[7,8], Siwei Chen[7,8], Paolo Sassone-Corsi[6], Pierre Baldi[7,8], Dina P. Matheos[1,2] & Marcelo A. Wood[1,2,3]

Aging is accompanied by impairments in both circadian rhythmicity and long-term memory. Although it is clear that memory performance is affected by circadian cycling, it is unknown whether age-related disruption of the circadian clock causes impaired hippocampal memory. Here, we show that the repressive histone deacetylase HDAC3 restricts long-term memory, synaptic plasticity, and experience-induced expression of the circadian gene *Per1* in the aging hippocampus without affecting rhythmic circadian activity patterns. We also demonstrate that hippocampal *Per1* is critical for long-term memory formation. Together, our data challenge the traditional idea that alterations in the core circadian clock drive circadian-related changes in memory formation and instead argue for a more autonomous role for circadian clock gene function in hippocampal cells to gate the likelihood of long-term memory formation.

[1] Department of Neurobiology and Behavior, University of California, Irvine, CA 92697, USA. [2] Center for the Neurobiology of Learning and Memory, Irvine, CA 92697, USA. [3] Institute for Memory Impairments and Neurological Disorders, University of California, Irvine, CA 92697, USA. [4] Department of Medical Microbiology and Immunology, University of California, Davis, CA 95616, USA. [5] Department of Biological Sciences, Mount Holyoke College, South Hadley, MA 01075, USA. [6] Center for Epigenetics and Metabolism, Department of Biological Chemistry, University of California, Irvine, CA 92697, USA. [7] Department of Computer Science, University of California, Irvine, CA 92697, USA. [8] Institute for Genomics and Bioinformatics, University of California, Irvine, CA 92697, USA. Correspondence and requests for materials should be addressed to M.A.W. (email: mwood@uci.edu)

Animals have an internal circadian clock that drives the rhythmic cycling of biological processes every ~24 h. Circadian rhythms drive numerous physiological events, including the sleep-wake cycle, feeding behavior, body temperature, and metabolism. In the master circadian clock, the suprachiasmatic nucleus (SCN), a group of core clock genes oscillate in a negative feedback loop that cycles every ~24 h[1,2]. In addition to regulating basic biological processes, the circadian clock also has a strong influence on memory. Long-term memory, which is transcription-dependent, shows a strong time-of-day effect, with peak memory performance during the day (inactive phase) in mice[3,4]. Notably, both long-term memory and circadian rhythmicity are impaired with age[5], suggesting that common molecular mechanisms might underlie both processes. One idea is that clock genes located in memory-relevant structures, like the dorsal hippocampus, might gate an animal's ability to form long-term memory based on the time of day[6]. Consistent with this, disruption of several individual clock genes throughout the brain can impair hippocampal long-term memory in young animals. As no study to date has selectively disrupted circadian gene function within the dorsal hippocampus, it is unclear whether clock genes act within hippocampal cells to affect long-term memory formation or whether these memory deficits result from off-target effects in other brain regions, such as impaired circadian rhythms, sleep deficits, or even developmental abnormalities.

Gene expression is decreased in the aging brain, which may be the consequence of a more repressive chromatin structure. Transcription is controlled in part through changes in chromatin structure, which can dynamically promote or restrict access to neuronal DNA following a learning event. One hypothesis put forth by Barnes and Sweatt posits that the epigenome is altered in aging neurons, resulting in a repressive chromatin structure that prevents normal gene expression required for long-term memory formation[7]. Several studies support this idea showing altered histone modification mechanisms in the aging brain[8–10]. However, whether chromatin modification mechanisms abnormally regulate circadian gene expression in a key learning and memory brain region is unknown. Here, we examined this possibility by focusing on the role of histone deacetylase 3 (HDAC3)-dependent regulation of age-related memory and gene expression. We found that deletion or disruption of HDAC3 in the dorsal hippocampus ameliorates age-related impairments in long-term memory and synaptic plasticity in 18-month-old mice, an effect that appears to be mediated in part by the core circadian clock gene Period1 (Per1). Broadly, this work suggests that Per1 may be a mechanism contributing to age-related impairments in both long-term memory and circadian rhythmicity, depending on the structure.

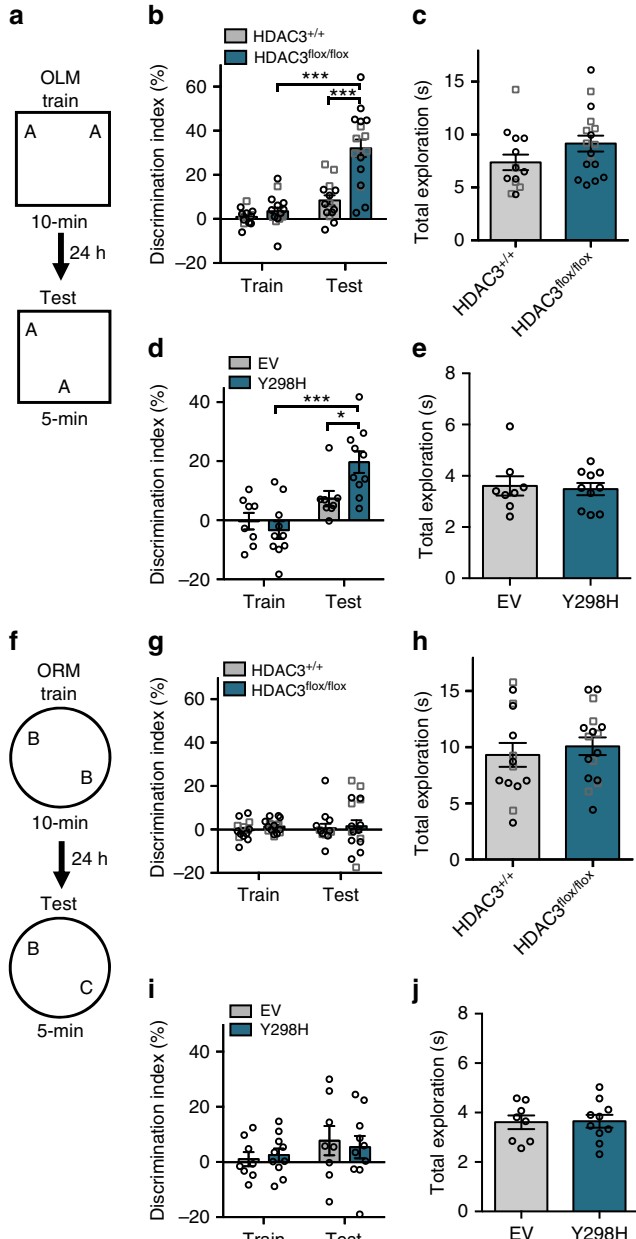

## Results

**HDAC3 contributes to age-related memory impairments**. We first tested whether the repressive histone deacetylase HDAC3 plays a role in age-related memory decline. HDAC3 is a potent negative regulator of memory formation and disruption of HDAC3 in young animals can transform a subthreshold learning event into one generating persistent long-term memory for multiple tasks[11–14]. We used two methods of disrupting HDAC3 in the dorsal hippocampus of aging (18-month-old) mice. First, we created focal homozygous deletions of HDAC3 by infusing AAV2.1-CaMKII-Cre recombinase (1 μL per side) into the dorsal hippocampi of HDAC3[flox/flox] mice (Supplementary Fig. 1a). Second, to selectively block the enzymatic activity of HDAC3, we used a dominant-negative point mutant virus, AAV2.1-CMV-HDAC3(Y298H)-v5 that specifically blocks HDAC3 deacetylase activity without affecting protein-protein interactions (see ref. [12,15–17] Supplementary Fig. 1b). Viruses were infused two

weeks before training, allowing for tight spatial and temporal control over our HDAC3 manipulations to avoid potential side effects that might occur from prolonged HDAC3 disruption during development[18,19]. Two weeks after AAV-CaMKII-Cre infusion (Supplementary Fig. 1c), we observed that Hdac3 mRNA expression was not affected by training in object location memory (OLM), but genetic deletion of hippocampal Hdac3 disrupted expression of Hdac3 mRNA (Supplementary Fig. 1d) in addition to HDAC3 protein (Supplementary Fig. 1a).

To determine whether deletion of HDAC3 improves memory in aging mice, we tested the effects of hippocampal HDAC3 deletion (HDAC3[flox/flox]) or activity disruption (HDAC3 (Y298H)) on long-term memory for OLM (Fig. 1a). Consistent with numerous reports of age-related hippocampal memory deficits, we found that aging, 18-m.o. wildtype (HDAC3[+/+]) mice displayed poor memory for OLM following 10-min training, showing no significant increase in DI between training and testing (Fig. 1b, gray bars). Importantly, a 10-min training session

**Fig. 1** Deleting or disrupting HDAC3 ameliorates age-related impairments in hippocampal memory. **a** OLM procedure. AAV was infused 2 weeks before training. **b** 18-m.o. HDAC3[flox/flox] mice showed significantly better memory for OLM compared to wild type (HDAC3[+/+]) littermates (Two-way ANOVA: Significant Genotype x Session interaction ($F_{(1,29)} = 15.96$, $p < 0.001$), Sidak's post-hoc tests, $***p < 0.001$, $n = 14(5F)$, $17(6F)$). **c** Total exploration was similar for both groups at test ($t_{(29)} = 1.67$, $***p = 0.11$). **d** Disrupting HDAC3 activity in the dorsal hippocampus with AAV-HDAC3 (Y298H)-V5 also ameliorated hippocampal memory impairments in 18-m.o. mice (Two-way ANOVA: main effect of session ($F_{(1,16)} = 15.96$, $p < 0.001$), Sidak's post hoc tests, $***p < 0.001$, $*p < 0.05$, $n = 9,10$; all males). **e** Total exploration time was similar for both groups at test ($t_{(16)} = 0.28$, $p = 0.78$). **f** ORM experimental procedure, 2 weeks after the completion of OLM. **g** Both 18-m.o. HDAC3[flox/flox] mice and HDAC3[+/+] littermates showed little preference for the novel object (Two-way ANOVA, no main effects or interaction, $n = 14(5 F)$, $17(6 F)$). **h** Total exploration time was similar for both groups at test ($t_{(29)} = 0.59$, $p = 0.56$). **i** Disrupting HDAC3 activity in the dorsal hippocampus with AAV-HDAC3(Y298H) also had no effect on ORM, with neither group showing preference for the novel object (Two-way ANOVA, no main effects or interaction, $n = 9,10$; all males). **j** Groups showed similar total exploration time at test ($t_{(16)} = 0.28$, $p = 0.78$). Data are presented as mean ± SEM; black circles, males; gray squares, females

normally produces strong long-term memory in young mice[20]. In contrast, 18-m.o. HDAC3[flox/flox] littermates formed robust long-term memory (Fig. 1b, teal bars), with a significant increase in preference for the moved object at test relative to training. HDAC3[flox/flox] mice showed a significantly higher DI at test than HDAC3[+/+] mice despite similar levels of total exploration during the test session (Fig. 1c). We observed similar effects with the activity-specific AAV-HDAC3(Y298H) virus. Aging, 18-m.o. empty vector (EV) control mice showed poor memory for OLM whereas mice infused with AAV-HDAC3(Y298H) into the DH showed significantly higher preference for the moved object with no change in total exploration (Fig. 1d, e). In contrast to the poor long-term memory observed in 18-m.o. wildtype mice (Fig. 1b, d), short-term memory for OLM (tested 60 m after acquisition; Supplementary Fig. 2a) was intact for both HDAC3[+/+] and HDAC3[flox/flox] mice (Supplementary Fig. 2b, c) and there was no significant difference in anxiety between these groups (Supplementary Fig. 2d). We also observed no significant difference in movement between the groups during habituation (Supplementary Figs. 2e, f, 8a, b). Together, these results demonstrate that age-related impairments in OLM are ameliorated by deletion or disruption of HDAC3 in the dorsal hippocampus.

We next tested whether our focal HDAC3 manipulation affected object recognition memory (ORM), which does not require the dorsal hippocampus for retrieval[20]. In this task, one familiar object is replaced by a novel item (Fig. 1f). Deleting HDAC3 in the dorsal hippocampus did not rescue memory for ORM, with both HDAC3[+/+] and HDAC3[flox/flox] mice showing no preference for the novel object at test compared to training (Fig. 1g). Again, groups did not differ in total exploration levels during the test session (Fig. 1h). Similarly, activity-specific disruption of HDAC3 in the hippocampus was unable to ameliorate age-related ORM impairments (Fig. 1i, j). Thus, age-related impairments in long-term ORM were not ameliorated by hippocampal deletion or disruption of HDAC3.

Together, our results indicate that deletion or disruption of HDAC3 in the dorsal hippocampus can ameliorate age-related long-term memory deficits for a hippocampus-dependent task (OLM; Fig. 1a–e) without affecting memory for a hippocampus-independent task (ORM; Fig. 1f–j). Importantly, all mice showed

intact short-term memory for OLM (Supplementary Fig. 2), suggesting that these animals acquire memory normally but fail to consolidate this information into observable long-term memory. As short-term memory is transcription-independent (for review[21]), this finding is consistent with the idea that learning-induced gene expression is altered in aging mice, resulting in age-related impairments in long-term memory.

**HDAC3 disruption reverses age-related impairments in LTP.** To test whether HDAC3 also contributes to age-related synaptic plasticity impairments, we examined long-term potentiation (LTP) in acute hippocampal slices following either deletion or disruption of HDAC3. LTP is also impaired with age, particularly when the stimulation protocol is close to the LTP induction threshold[22]. Two weeks after viral infusion, we prepared hippocampal slices and induced LTP with a single train of 5 theta bursts to Schaffer collateral inputs and recorded field EPSPs from apical dendrites of CA1b. This relatively mild form of stimulation produced a stable level of LTP in young HDAC3[+/+] slices (Fig. 2a). Deleting HDAC3 in the hippocampus enhanced LTP, with HDAC3[flox/flox] mice showing significantly higher potentiation than wildtype controls. As predicted, aging-18-m.o. HDAC3[+/+] mice showed impaired LTP and the HDAC3 deletion ameliorated this deficit, producing LTP comparable to that of young wildtype mice (Fig. 2b, c) with no effect on baseline synaptic transmission (Fig. 2g, h).

We observed similar results with the activity-specific disruption. Here, we used a within-subjects design in which young and old wildtype mice were infused with the control virus (AAV-EV) into one hippocampus and AAV-HDAC3(Y298H) into the contralateral hippocampus. As before, we found that disrupting HDAC3 activity enhanced LTP in slices from young mice (Fig. 2d) and ameliorated age-related LTP impairments in slices from aging mice (Fig. 2e, f) without interfering with baseline synaptic transmission (Fig. 2i, j). Either deletion or disruption of HDAC3 can therefore ameliorate age-related impairments in hippocampal LTP.

**A subset of age-impaired genes is improved by HDAC3 deletion.** Our data suggest that deleting or disrupting HDAC3 ameliorates age-related impairments in long-term memory and synaptic plasticity. We next asked whether age-related deficits in hippocampal gene expression could also be ameliorated by deleting HDAC3. We hypothesized that dysregulation of HDAC3 activity in the old brain leads to an unusually repressive chromatin structure that limits gene expression, which ultimately impairs long-term memory. To identify which specific genes are regulated by HDAC3 in the young and aging brain, we ran an RNA sequencing experiment in which we compared young (3-m.o.) wildtype mice, aging (18-m.o.) HDAC3[+/+] mice, and aging (18-m.o.) HDAC3[flox/flox] littermates. To identify gene expression changes during memory consolidation, animals in each group were killed 60 m after 10-min OLM training and compared to homecage (HC) controls. After mapping and considering the haploid genome[23,24], sequencing quality was assessed (Supplementary Fig. 3a, b) and significant differences in expression profiles were examined between all pairs of samples for $p < 0.05$[20].

We expected that experience-induced gene expression would be altered in the old brain, as previously described, with a subset of genes failing to express after learning. We therefore focused on those genes expressed at significantly higher levels in the trained groups compared to homecage controls. While each group (Young WT, Old WT, Old HDAC3[flox/flox]) showed a substantial number of genes induced by OLM training, each group showed a unique gene expression profile (Fig. 3a, Supplementary Tables 1–3). Old brains

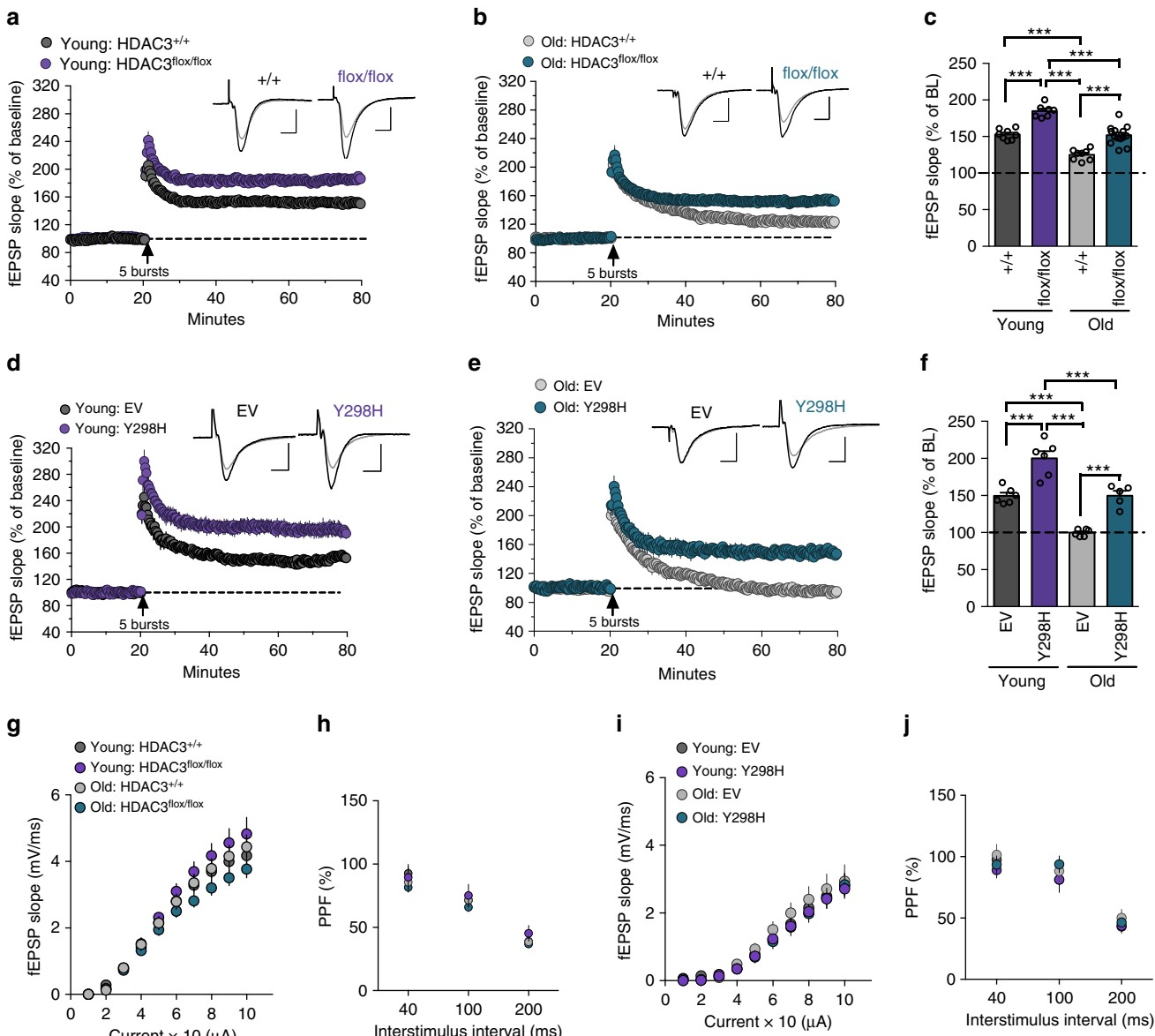

**Fig. 2** Deleting or disrupting HDAC3 ameliorates age-related impairments in synaptic plasticity. **a** Mean ± SEM fEPSP slope recordings in hippocampal slices from young (3-m.o.) HDAC3[flox/flox] mice or HDAC3[+/+] littermates 2-weeks after hippocampal AAV-Cre infusion. Deleting HDAC3 enhanced theta burst-induced LTP in the young hippocampus. Scale bar: 1 mV per 5 ms. **b** The same stimulation caused a gradual decay toward baseline in slices from old (18-m.o.) HDAC3[+/+] mice. Deleting HDAC3 (HDAC3[flox/flox]) restored a greater level of stable potentiation relative to HDAC3[+/+] littermates. Scale bar: 1 mV per 5 ms. **c** Summary graph showing mean fEPSP slope 60 m after stimulation. Potentiation was significantly enhanced by HDAC3 deletion in slices from both young and old mice. Deleting HDAC3 in the 18-m.o. hippocampus produced LTP comparable to that of 3-m.o. wildtype mice (Two-way ANOVA: main effects of Age ($F_{(1,33)} = 80.8$, $p < 0.0001$) and Genotype ($F_{(1,33)} = 75.5$, $p < 0.0001$), Sidak's post hoc tests, ***$p < 0.0001$, $n = 8, 7, 8, 14$ slices from 3, 3, 3, 6 mice; all male). **d** Mean ± SEM fEPSP slope recordings in hippocampal slices from young (3-m.o.) mice two weeks after hippocampal AAV-HDAC3 (Y298H) or AAV-EV infusion. Disrupting HDAC3 activity enhanced LTP in the young hippocampus. Scale bar: 1 mV per 5 ms. **e** The same stimulation protocol failed to produce stable potentiation in slices from 18-m.o. mice, but this impairment was overcome by disrupting HDAC3 activity (AAV-HDAC3 (Y298H)). Scale bar: 1 mV per 5 ms. **f** Summary graph. Potentiation was significantly enhanced by HDAC3 disruption in slices from both young and old mice. Disrupting HDAC3 activity in the 18-m.o. hippocampus with AAV-HDAC3(Y298H) produced LTP comparable to that of 3-m.o. wildtype mice (Two-way ANOVA: main effects of Age ($F_{(1,19)} = 62.8$, $p < 0.0001$) and Genotype ($F_{(1,19)} = 63.4$, $p < 0.0001$), Sidak's post hoc tests, ***$p < 0.0001$, $n = 6, 6, 6, 5$ slices from 4, 4, 4, 4 mice; all male). **g** Input/output (I/O) curves and **h** paired-pulse facilitation (PPF) were comparable between HDAC3[+/+] and HDAC3[flox/flox] slices. **i** I/O curves and **j** PPF were similar between AAV-EV and AAV-HDAC3(Y298H) slices. Data are presented as mean ± SEM

(both wildtype and HDAC3[flox/flox]) showed a greater number of upregulated and downregulated genes in response to training than young wildtype brains (Fig. 3b), with a bias towards downregulation (Fig. 3c) as previously observed[25]. Most of the genes upregulated by experience were unique to that group, with little overlap between the three conditions (Fig. 3d, e). This indicates that the aging brain is characterized by both failed induction of genes that are typically

upregulated by learning *and* aberrant induction of genes that are not usually expressed during memory consolidation. Further, when HDAC3 is deleted from the aging brain, OLM training induces a unique gene expression profile, rather than recapitulating the gene expression profile of the young brain.

Only a handful of genes were common to two or more groups (Fig. 3d, e). Of particular interest is the subset of genes

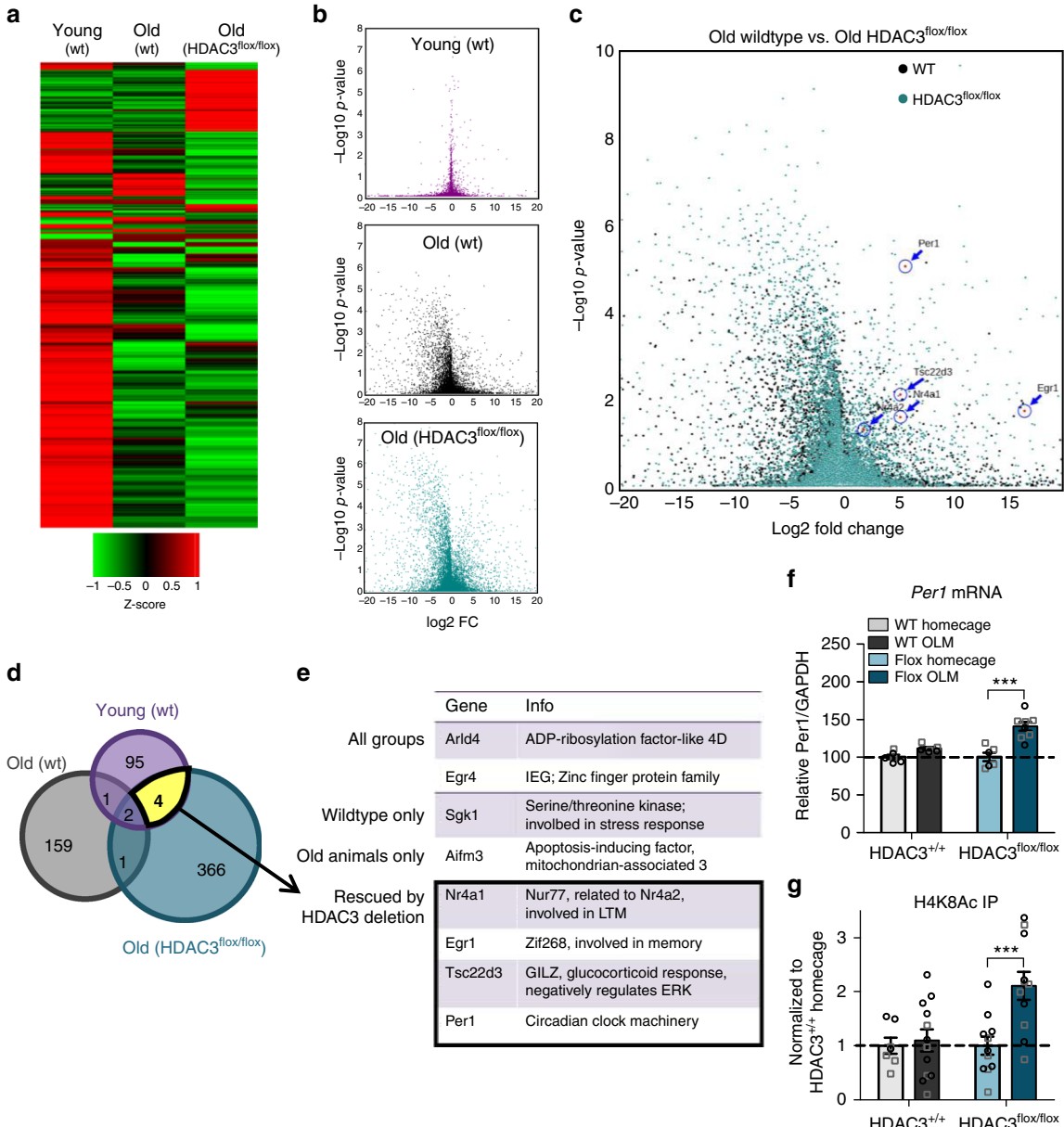

**Fig. 3** A subset of experience-induced genes is impaired with age and rescued by HDAC3 deletion. **a** Heat map comparing experience-induced gene changes in each group (Young wt: HC $n = 6$(3F), OLM $n = 6$(3F); Old wt: HC $n = 6$(2F), OLM $n = 6$(2F); Old HDAC3$^{flox/flox}$: HC $n = 8$(5F), OLM $n = 8$ (5F)). Each row represents an individual gene. Color legend shown at bottom; red, upregulation; green, downregulation. **b** Volcano plots illustrating the significance ($Y$-axis) and magnitude ($X$-axis) of experience-induced changes in each group. Young animals showed fewer experience-induced changes in gene expression compared to old wildtype or old HDAC3$^{flox/flox}$ mice. **c** Overlapping volcano plot comparing old wildtype and old HDAC3$^{flox/flox}$ groups. *Per1* was strongly induced by OLM training in HDAC3$^{flox/flox}$ mice. **d** Gene expression diagram displaying genes expressed higher in trained animals compared to homecage controls. Total gene count shown inside circles. **e** Table listing experience-induced genes common to two or more groups. Of particular interest are the four genes "rescued by HDAC3 deletion" that are upregulated in the young wildtype and old HDAC3$^{flox/flox}$ groups but are not upregulated in the old wildtype group. **f** *Per1* mRNA expression is increased in the 18-m.o. brain in the absence of HDAC3 (Two-way ANOVA: Genotype x Train ($F_{(1,22)} = 9.48$, $p < 0.01$), Sidak's post hoc tests, ***$p < 0.001$; $n = 6$(2F), 6(3F), 6(4F), 8(5F)). **g** Occupancy of H4K8Ac at the *Per1* promoter is upregulated by OLM training in the 18-m.o. hippocampus in the absence of HDAC3 (Two-way ANOVA: Genotype x Train ($F_{(1,37)} = 5.5$, $p < 0.05$), Sidak's post hoc tests, ***$p < 0.001$, $n = 7$(4F), 12(4F), 11(4F), 11(6F)). Data are presented as mean ± SEM; black circles, males; gray squares, females

upregulated in both the young wildtype and old HDAC3$^{flox/flox}$ groups that do not show experience-induced increases in the old wildtype hippocampus. These are the genes that fail to normally express in the old brain after OLM training but are rescued by HDAC3 deletion and may therefore function as a mechanism through which HDAC3 deletion ameliorates age-related memory impairments. Four genes were identified in this group: *Nr4a1*, *Egr1*, *Tsc22d3*, and *Per1*. All of these genes have

been broadly implicated in memory formation[6,26,27], although this is the first study to demonstrate that experience-induced expression of these genes is impaired with age and rescued with HDAC3 deletion.

**Deleting HDAC3 ameliorates age-related deficits in *Per1*.** Of the genes identified through our RNA-seq, Period1 (*Per1*) was the

most strongly induced by OLM training in the HDAC3$^{flox/flox}$ group (Fig. 3c). This target is particularly intriguing, as Per1 is typically studied in the context of circadian rhythms but has also been implicated in hippocampal memory formation[6,28,29]. As aging is known to be accompanied by impairments in circadian rhythms[5] and memory is linked to time-of-day[2,30], this target of HDAC3 may represent a critical and underexplored interface between aging, chromatin modification, and the circadian clock.

To further examine the expression of Per1 in old HDAC3$^{+/+}$ and old HDAC3$^{flox/flox}$ animals, we used RT-qPCR and ChIP-qPCR. We found that OLM training failed to induce upregulation of Per1 in the dorsal hippocampus of 18-m.o. HDAC3$^{+/+}$ mice, but in the absence of HDAC3 (HDAC3$^{flox/flox}$), OLM training triggered a significant increase in Per1 mRNA (Fig. 3f). We also measured the expression of two additional genes that play a well-documented role in long-term memory formation: Arc and cFos[31]. Experience-induced expression of Arc was intact in the aging wildtype brain and was unaffected by HDAC3 deletion (Supplementary Fig. 3c). cFos expression, on the other hand, failed to be induced by OLM training in the old HDAC3$^{+/+}$ hippocampus, but deleting HDAC3 was not sufficient to restore this failed induction (Supplementary Fig. 3d). Deleting HDAC3 therefore only restores expression of a subset of experience-induced genes in the aging brain, including Per1.

To determine whether deleting HDAC3 restores expression of Per1 by promoting histone acetylation along its promoter, we next measured acetylation of histone 4, lysine 8 (H4K8Ac) at the Per1 CRE promoter site using chromatin immunoprecipitation (ChIP-qPCR). H4K8Ac is a marker of transcriptional activation[32] and is thought to be a target of HDAC3[11,12]. OLM training did not change H4K8Ac levels at the Per1 promoter in the old wildtype brain but in the absence of HDAC3, H4K8Ac levels at the Per1 promoter were significantly increased in response to OLM training (Fig. 3g). For Arc and cFos, we saw no change H4K8Ac occupancy across groups (Supplementary Fig. 3e, f). Together, these results suggest that deleting HDAC3 increases acetylation at the Per1 promoter and expression of Per1 mRNA in response to learning.

One important question is whether the observed changes in Per1 are due to changes in the circadian rhythm of HDAC3$^{flox/flox}$ mice. If deletion of HDAC3 in the dorsal hippocampus alters circadian rhythmicity, this could explain the observed changes in both Per1 expression and long-term memory formation. To rule this out, we assessed the circadian rhythmicity of young (3-m.o.) and aging (18-m.o.) HDAC3$^{flox/flox}$ mice and their HDAC3$^{+/+}$ littermates following AAV-CaMKII-Cre infusion. After 2 weeks of entrainment to a 12 h light/dark cycle (LD), mice were put in constant darkness (DD) to measure endogenous circadian rhythms (Supplementary Fig. 4a). We observed no difference in circadian activity patterns between HDAC3$^{+/+}$ and HDAC3$^{flox/flox}$ mice at either age group (Supplementary Fig. 4b–f), suggesting that hippocampal HDAC3 has no effect on circadian rhythmicity. Thus, the observed changes in Per1 expression and long-term memory formation following HDAC3 deletion cannot be explained by changes in circadian rhythmicity.

**Per1 is induced by OLM training and regulated by HDAC3.** We next wanted to determine whether hippocampal Per1 is required for long-term memory formation. First, we assessed whether hippocampus-dependent learning typically induces Per1 mRNA expression. We sacrificed young (3-m.o.) wildtype mice 60 m after acquisition of either OLM or context fear conditioning (CFC)[12]. Per1 mRNA expression was significantly upregulated in animals trained with either OLM (Fig. 4a) or CFC (Fig. 4b) compared to homecage controls, indicating that Per1 mRNA

expression is typically induced in the hippocampus during memory consolidation for multiple tasks.

To determine whether this experience-induced increase in Per1 might be mediated through HDAC3, we next used ChIP-qPCR to measure HDAC3 occupancy after OLM training at different sites along the Per1 promoter in the young hippocampus (Fig. 4c, top). We found that HDAC3 occupancy at the Per1 promoter was reduced at all three tested sites following OLM training (Fig. 4c, bottom). Along with our previous finding that HDAC3 deletion restores both acetylation at Per1 and Per1 mRNA expression (Fig. 3), this strongly suggests that HDAC3 regulates Per1 expression in the dorsal hippocampus and dysregulation of HDAC3 may contribute to age-related impairments in experience-induced Per1 expression. PER1 may therefore be part of a key mechanism through which HDAC3 deletion ameliorates age-related impairments in hippocampal long-term memory formation.

**Knockdown of Per1 impairs long-term memory in young mice.** Next, to test whether upregulation of Per1 in the dorsal hippocampus is necessary for long-term memory formation, we infused siRNA targeting Per1 into the dorsal hippocampus of young mice 48 h before 10-min OLM training. Infusion of Per1 siRNA produced a significant reduction of PER1 protein in the dorsal hippocampus, as measured 2 h after the final test session (Fig. 4d, Supplementary Fig. 5). This relatively modest knockdown of PER1 protein (~30%) produced severely impaired memory formation for OLM; mice infused with Per1 siRNA showed no significant increase in DI between training and testing and displayed significantly less preference for the moved object during testing than control mice (Fig. 4e) with no effect on total object exploration at test (Fig. 4f) and no difference in movement during the pre-training habituation sessions (Supplementary Fig. 8c). This demonstrates, for the first time, that local disruption of a core circadian clock gene selectively within the hippocampus can impair memory formation.

**Per1 overexpression improves memory in aging mice.** Finally, to determine whether overexpression of Per1 in the dorsal hippocampus is sufficient to ameliorate age-related memory impairments, we locally upregulated Per1 using two complementary methods. First, we used a lentivirus expressing wild-type Per1 with a v5 epitope tag (pLVX-v5Per1) (Fig. 5a, Supplementary Fig. 6a). To confirm that this plasmid over-expresses Per1, we transfected HT22 cells with pLVX-v5Per1 or pLVX-EV and measured Per1 mRNA expression. At both 24 and 48 h after transfection, Per1 mRNA was significantly increased in cells transfected with pLVX-v5Per1 relative to cells transfected with the control plasmid (Fig. 5b). Transfection of pLVX-v5Per1 also led to a significant increase in mRNA for Per2 (a Period clock family member that interacts with Per1) at 24 h (Supplementary Fig. 6b), although this increase was far smaller than the observed increase in Per1 (at 24 h, Per1: 466-fold increase over EV, Per2: 1.6-fold increase over EV). Per1 overexpression did not affect the transcription of the nearest downstream gene Hes7, however (Supplementary Fig. 6c).

To determine whether Per1 overexpression improves memory performance in aging mice, pLVX-v5Per1 was infused into the dorsal hippocampus of 18-m.o. mice 2 weeks before behavior. pLVX-v5Per1 mice showed significantly improved memory for OLM at test relative to pLVX-EV (empty vector) controls (Fig. 5c). Only pLVX-v5Per1 mice showed a significant increase in preference for the moved object at test relative to training with no observed group differences in total exploration at test (Fig. 5d) or in movement during habituation (Supplementary Fig. 8d).

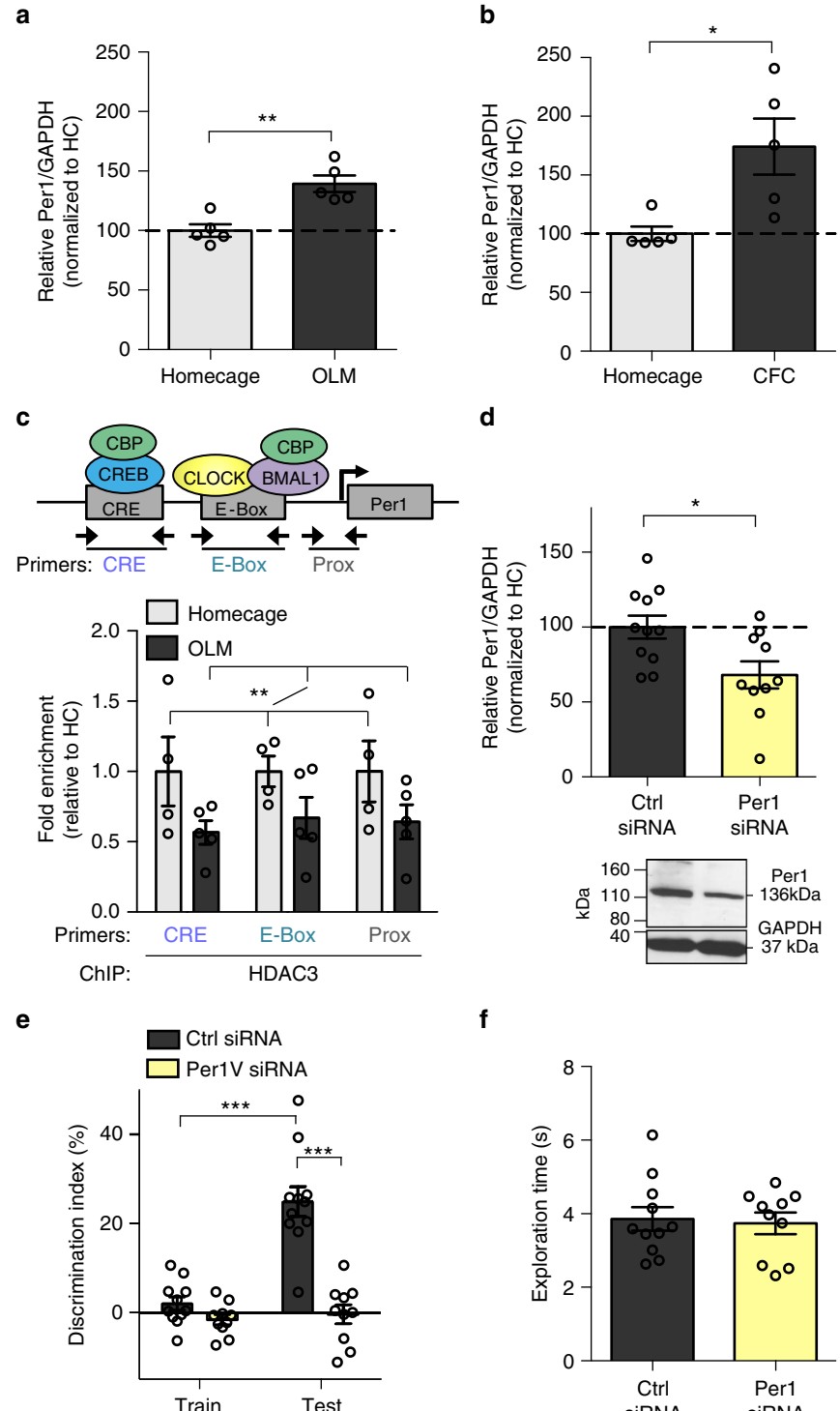

**Fig. 4** Knockdown of PER1 in the dorsal hippocampus impairs long-term memory. *Per1* mRNA expression is upregulated 60 m in the young hippocampus after **a** OLM ($t_{(8)} = 4.49$, **$p < 0.01$; $n = 5, 5$; all male) and **b** context fear conditioning ($t_{(8)} = 3.01$, *$p < 0.05$, $n = 5, 5$; all male). **c** Top, schematic of ChIP primer sites along the *Per1* promoter. Bottom, HDAC3 ChIP results. HDAC3 occupancy was reduced along the *Per1* promoter in the young hippocampus after OLM (Training effect only, $F_{(1,21)} = 8.51$, **$p < 0.01$, $n = 4, 5$; all male). **d** Hippocampal PER1 protein expression was significantly reduced by *Per1* siRNA infusion ($t_{(20)} = 2.72$, *$p = 0.01$, $n = 11,11$; all male). **e** *Per1* knockdown impaired OLM (Two-way ANOVA: siRNA x Session ($F_{(1,19)} = 29.22$, $p < 0.001$), Sidak's post hoc tests, ***$p < 0.001$, $n = 11, 10$; all male). **f** *Per1* siRNA did not affect total exploration time at test ($t_{(19)} = 0.27$, $p = 0.79$). Data are presented as mean ± SEM

Following behavior, hippocampal tissue from a subset of animals was processed for RNA sequencing to confirm the presence of the pLVX-EV and pLVX-v5Per1 transcripts in the appropriate groups in vivo (Supplementary Fig. 7a, b, f, g).

To complement this approach, we also used the CRISPR/dCas9 Synergistic Activation Mediator (SAM) system[33] to drive transcriptional activation of *Per1* in the dorsal hippocampus. This system consists of three lentiviral components: a catalytically

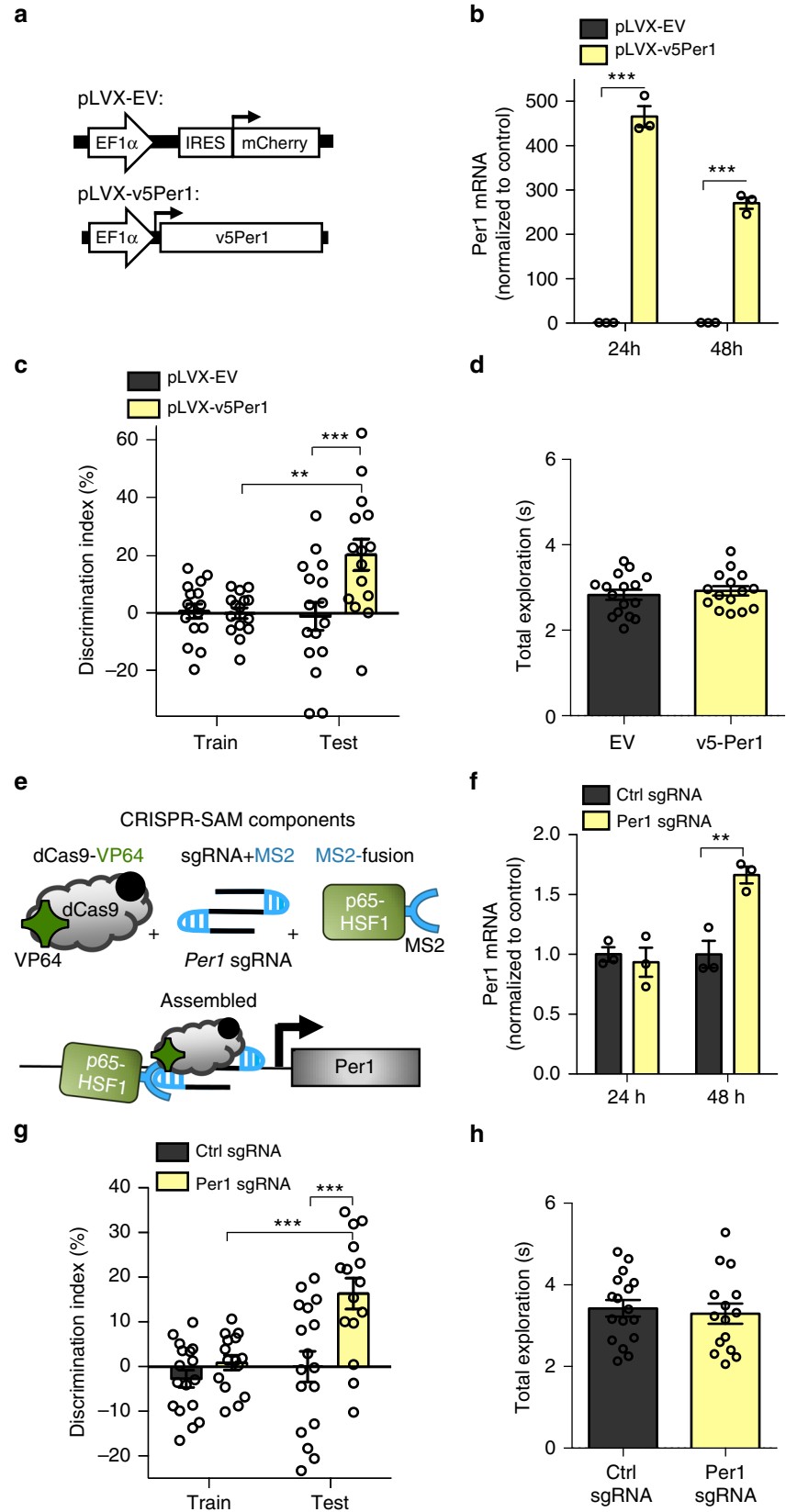

inactive Cas9 (dCas9) fused to a VP64 transcriptional activation domain with a GFP tag (dCas9-VP64-GFP), a modified single-guide RNA (sgRNA) targeting *Per1* with two MS2 RNA adaptamers that can recruit the third component, an MS2-double transcriptional activator (MS2-p65-HSF1) fusion protein

(Fig. 5e). Control animals received a control sgRNA lacking the 20 nucleotide sequence required to target the SAM system to *Per1* (referred to as ctrl sgRNA). Assembly of these components at the *Per1* promoter allows the three effector domains (VP64, p65, and HSF1) to drive *Per1* transcription. To confirm the effectiveness of

**Fig. 5** Overexpression of *Per1* in the dorsal hippocampus ameliorates age-related impairments in object location memory. **a** Schematic of lentivirus construct used to overexpress v5-tagged *Per1* (pLVX-v5Per1) compared to the empty vector control (pLVX-EV). **b** *Per1* mRNA was significantly increased 24 and 48 h after transfection of pLVX-v5Per1 in HT22 cells compared to cells transfected with EV (Two-way ANOVA: Group x Timepoint interaction ($F_{(1,8)} = 52.8$, $p < 0.001$), Sidak's post hoc tests, ***$p < 0.001$, $n = 3, 3, 3, 3$). **c** 18-m.o. mice given hippocampal infusions of pLVX-v5Per1 showed significantly better memory for OLM than mice given pLVX-EV control virus (Two-way ANOVA: virus x session interaction, ($F_{(1,29)} = 7.15$, $p < 0.05$), Sidak's post hoc tests, **$p < 0.01$, ***$p < 0.001$, $n = 16, 15$, all males). **d** Total exploration was similar for both groups at test ($t_{(29)} = 0.57$, $p = 0.57$). **e** Schematic of CRISPR/dCas9 Synergistic Activation Mediator (SAM) system used to drive *Per1* transcription. Top: Individual components of SAM. Bottom: Components assembled at the *Per1* promoter, driving *Per1* transcription. **f** *Per1* mRNA was significantly increased 48 h after transfection of CRISPR-SAM components in HT22 cells compared to cells transfected with the control sgRNA (Two-way ANOVA: Group x Timepoint interaction ($F_{(1,8)} = 14.77$, $p < 0.01$), Sidak's post hoc tests, **$p < 0.001$, $n = 3,3,3,3$). **g** 18-m.o. mice given hippocampal infusions of the CRISPR-SAM system with sgRNA targeting *Per1* showed significantly better memory for OLM compared to EV control mice with control sgRNA (Two-way ANOVA: Virus x Session interaction ($F_{(1,30)} = 5.83$, $p < 0.05$), Sidak's post hoc tests, ***$p < 0.001$, $n = 17, 15$, all males). **h** Total exploration was similar for both groups at test ($t_{(30)} = 0.41$, $p = 0.68$). Data are presented as mean ± SEM

the CRISPR-SAM system, we transfected HT22 cells with all three plasmids, harvested the cells 24 or 48 h later, and measured *Per1* mRNA expression. By 48 h after transfection, *Per1* mRNA was significantly increased in the group given *Per1* sgRNA compared to controls (Fig. 5f), confirming that the CRISPR-SAM system effectively drives *Per1* mRNA expression. We observed no change in expression for either *Per2* mRNA (Supplementary Fig. 6e) or *Hes7* mRNA (Supplementary Fig. 6f), indicating that the CRISPR-SAM system drives selective overexpression of the target gene, *Per1*.

Next, 18-m.o. mice were given intra-hippocampal infusions of the CRISPR-SAM lentiviruses (Supplementary Fig. 6d) and trained in OLM 2 weeks later. As observed with our pLVX-v5Per1 overexpression, CRISPR-SAM-mediated *Per1* overexpression significantly improved memory performance in *Per1* sgRNA-infused mice relative to control animals (Fig. 5g) without affecting total exploration (Fig. 5h) or movement during habituation (Supplementary Fig. 8e). Only mice given the *Per1* sgRNA showed a significant increase in preference for the moved object at test relative to training, indicating intact memory for the object locations (Fig. 5g). Following behavior, hippocampal tissue from a subset of animals was processed for RNA sequencing to confirm the presence of the CRISPR transcripts in the appropriate groups in vivo (Supplementary Fig. 7c–g). Together, these results demonstrate that overexpression of *Per1* in the dorsal hippocampus is sufficient to ameliorate age-related impairments in long-term object location memory. *Per1* is therefore a key gene that is critical for long-term memory formation, is regulated by HDAC3, and is impaired in the aging brain.

## Discussion

Our results show that deletion or disruption of the repressive histone deacetylase HDAC3 can ameliorate age-related impairments in both long-term memory and synaptic plasticity. Further, deletion of HDAC3 restores experience-induced expression of the circadian gene *Per1* in the dorsal hippocampus. As hippocampal PER1 expression is critical for long-term memory formation (Fig. 4) and overexpression of *Per1* in the hippocampus ameliorates age-related memory impairments (Fig. 5), PER1 is a potential mechanism through which deletion of HDAC3 improves memory and synaptic plasticity in aging mice. More broadly, age-related disruption of *Per1* might connect age-related impairments in both long-term memory and circadian rhythmicity, depending on the structure.

One key finding from the current study was that age-related impairments in hippocampal LTP could be ameliorated with HDAC3 deletion or disruption. This is consistent with recent work from the Sajikumar lab demonstrating that pharmacological blockade of HDAC3 can also ameliorate age-related impairments

in associative hippocampal LTP[34]. Interestingly, we found that slices from old HDAC3[flox/flox] and HDAC3(Y298H) animals failed to reach the same level of potentiation as slices from young HDAC3[flox/flox] or HDAC3(Y298H) animals (Fig. 2c, f), suggesting that aging brains may have a lower plasticity ceiling than young brains. One possible explanation for this difference in potentiation is that age-related loss of synaptic contacts in CA1[22] could lower the plasticity ceiling in the aging hippocampus, as the fewer available synapses would become saturated more quickly than the relatively abundant synapses in the young DH. If this is the case, strengthening the stimulation protocol or providing spaced stimulation bouts[35] should not further enhance LTP in the aging hippocampus, as no additional synapses are available. Further work will be necessary to determine the mechanism underlying this difference.

Our RNA sequencing results demonstrated that only a small subset of genes fit the criteria of being restored in the aging brain by HDAC3 deletion (Fig. 3e). Thus, rather than recapitulating the gene expression profile of the young brain, deleting HDAC3 in the aging brain restored experience-induced expression of a few key genes that are critically important for long-term memory formation, including *Per1*. *Per1* appears to be directly regulated by HDAC3, as focal deletion of HDAC3 restored experience-induced *Per1* expression and HDAC3 is physically removed from the *Per1* promoter in response to OLM training. Further, *Per1* expression is necessary for memory, as siRNA-mediated knockdown of PER1 protein in the DH impaired long-term memory formation in young mice and overexpression improved memory for OLM in aging mice. Notably, both lentiviral systems used to overexpress *Per1* only transfected a small number of cells in the CA1b area of the dorsal hippocampus (Supplementary Fig. 6a, d), making it necessary to confirm the presence of these constructs with RNA-sequencing (see methods). As previous work has shown that this precise region of the dorsal hippocampus is critical for OLM[36], this demonstrates that even a small focal alteration of *Per1* within a memory-relevant structure can impact long-term memory formation. This extends previous research showing that nonspecific deletion of *Per1* throughout the brain can impair memory formation in young mice[6,28,29]. Abnormal HDAC3-mediated repression of *Per1* in the aging brain is therefore a key event that could lead to age-related impairments in both long-term memory formation and circadian rhythmicity.

Although this study clearly demonstrates that *Per1* is an important gene through which HDAC3 regulates long-term memory in the aging brain, other genes most likely contribute to the memory-enhancing effects of HDAC3 deletion. Other genes proposed to mediate the effects of HDAC3 on synaptic plasticity and memory include NF-κB[34], FMRP[37], and *Nr4a2*[26]. In the current study, we identified an additional three genes (Fig. 3f:

*Nr4a1, Egr1, Tsc22d3*) that, like *Per1*, show impaired expression in the aging brain that is improved by HDAC3 deletion. How these different genes uniquely contribute to the memory-enhancing effects of HDAC3 deletion is currently unclear. Notably, all of these genes interface with the CBP/CREB pathway, which is critical for long-term memory formation[38,39] and is both up- and downstream of PER1[6,28]. As PER1 is upstream of CREB phosphorylation[6], it is possible that HDAC3-mediated repression of *Per1* reduces CREB phosphorylation, ultimately impairing the transcription of CREB-mediated genes like *Fmrp*[40,41], and the *Nr4a* gene family members *Nr4a1* and *Nr4a2*[26]. Understanding the complex dynamics between HDAC3, PER1, and these other molecular players will be an important goal of future research.

In the current study, we used two complementary methods to overexpress *Per1* in the aging brain: overexpression of a full-length *Per1* cDNA construct using pLVX-v5Per1 and transcriptional activation of endogenous *Per1* using the CRISPR-SAM system. Although pLVX-mediated overexpression drove a large increase in *Per1* mRNA (Fig. 5b), this was accompanied by an increase in *Per2*, another Period family member (Supplementary Fig. 6b). The expression of the downstream gene *Hes7*, however, was unchanged by pLVX-v5Per1 (Supplementary Fig. 6c). The CRISPR-SAM system, on the other hand, produced a more subtle increase in *Per1* mRNA that avoided off-target enhancements of either *Per2* or *Hes7* expression (Supplementary Fig. 6e–f). As both approaches similarly improved long-term memory in aging mice, it is unlikely that the off-target increase in *Per2* observed with pLVX-v5Per1 was the cause of the observed improvement in long-term memory.

Circadian effects on long-term memory are traditionally believed to stem from dysregulation within the SCN, which then drives alterations in peripheral structures involved in memory formation, like the dorsal hippocampus. Little is known about the role of individual circadian clock genes in the DH, despite a clear connection between circadian rhythmicity and long-term memory formation. Memory is closely linked to time-of-day, as plasticity-related gene cascades show circadian oscillations[3,6] and memory can be acquired more easily at certain periods of the circadian cycle. For example, context fear conditioning is acquired more strongly during the daytime, when MAPK phosphorylation levels peak[3]. Further, it is well-documented that aging is accompanied by a breakdown of circadian rhythms, presumably due to changes in the central circadian clock, the suprachiasmatic nucleus (SCN) (for review[1]). How this disruption in the circadian clock relates to the age-related impairments in memory is an open question. Our results suggest that HDAC3 limits experience-induced Per1 in the aging hippocampus, possibly contributing to the observed impairments in long-term memory. Epigenetic repression of Per1 may therefore represent an important interface between age-related impairments in both circadian rhythmicity and long-term memory formation.

Of the core canonical circadian clock genes, *Per1* is uniquely poised to dramatically affect hippocampal long-term memory. *Per1* appears to be predominantly involved in SCN output pathways and plays a key role in peripheral clocks downstream of the SCN, such as the hippocampus[42]. Further, recent work suggests that *Per1* may "gate" spatial memory formation throughout the day/night cycle by controlling CREB phosphorylation[6,28,29]. Indeed, hippocampal *Per1* mRNA upregulation has been observed after context or spatial learning in at least three other RNA-seq studies, including work in rats, indicating that *Per1* is typically upregulated after learning across species[20,43,44]. Along with the results of the current study, this work indicates that PER1 expression is critically important for hippocampal long-term memory formation; reductions in PER1 that occur at night[29] or with aging could impair hippocampal memory. To date,

research implicating *Per1* in memory formation has exclusively relied on global knockouts that disrupt *Per1* expression in the core circadian clock and other regions in addition to memory-relevant structures like the hippocampus[6,28,29,45,46], making it impossible to determine whether hippocampal PER1 is specifically required for memory formation. Indeed, global *Per1* deletion does affect circadian rhythmicity in some reports[42]. Here, we show for the first time that reducing PER1 expression directly in the dorsal hippocampus can impair memory in young mice whereas local overexpression of *Per1* in the dorsal hippocampus can improve memory in aging mice. As selective deletion HDAC3 (which regulates *Per1*) in the dorsal hippocampus had no effect on circadian activity patterns in the current study (Supplementary Fig. 4), and even electrolytic lesions of the dorsal hippocampus are insufficient to affect circadian rhythmicity[47], it seems unlikely that manipulating *Per1* within the dorsal hippocampus affects the function of the central circadian clock. Nonetheless, it is possible that experience-induced increases in *Per1* or virus-mediated overexpression of *Per1* can affect the circadian oscillation of other molecular players, such as other clock genes, within hippocampal neurons, even without affecting circadian activity patterns. Understanding how *Per1* functions to alter memory formation, including identifying these potential interacting partners, will be the target of future work.

Together, these results demonstrate that the core circadian clock gene *Per1* plays a key role within local memory structures to alter memory formation, a role that is independent of its function in the SCN. More generally, this challenges the traditional hypothesis that circadian changes in memory formation are driven by alterations in the core circadian clock and instead supports the hypothesis that circadian clock genes play a more autonomous role in hippocampal cells, possibly gating memory formation based on the time of day[6].

## Methods

**Mice**. Young adult mice were between 2 and 4 months old at the time of testing and aging mice were between 18 and 20 months old. All mice were C57BL/6J or maintained on a C57BL/6 background (HDAC3[+/+] and HDAC3[flox/flox] mice). Mice had free access to food and water and lights were maintained on a 12 h light/dark cycle. All behavioral testing was performed during the light cycle. All experiments were conducted according to US National Institutes of Health guidelines for animal care and use and were approved by the Institutional Animal Care and Use Committee of the University of California, Irvine.

**Surgery**. Mice were anesthetized with isoflurane (induced, 4%; maintained 1.5–2.0%) and placed in the stereotax. Injection needles were lowered to the dorsal hippocampus at a rate of 0.2 mm/15 s (AP, −2.0 mm; ML, ± 1.5 mm, DV, −1.5 mm relative to Bregma). Two minutes after reaching the target depth, 1.0 μl of virus or siRNA was infused bilaterally into the DH at a rate of 6 μl/h. For the CRISPR-SAM lentiviral infusions, a cocktail of the three viruses was infused to a final volume of 1.5 μL per hemisphere at the same rate. Injection needles remained in place for two minutes post-injection to allow the virus to diffuse. The injectors were then raised 0.1 mm and allowed to sit for another minute before being removed at a rate of 0.1 mm per 15 s. Viral infusions were performed 2 weeks before behavioral analysis whereas siRNA knockdown was performed 2d before training[13]. For all injection experiments animals were randomly assigned to the different injection conditions (with the exception of HDAC3[+/+] and HDAC3[flox/flox] mice, which were all injected with AAV-CaMKII-Cre). For all behavioral experiments, animals within each viral condition were randomly assigned to homecage/trained groups and all conditions (objects, boxes, etc.) were counterbalanced between groups.

**AAV production**. AAV2.1-CaMKII-Cre was purchased from Penn Vector Core (titer: 1.81 × 10[13] GC/ml). For AAV2.1-HDAC3(Y298H)-v5, we amplified wildtype HDAC3 from hippocampal cDNA and cloned the product into a modified pAAV-IRES-hrGFP (Agilent) under control of the CMV promoter and β-globin intron. The 3×-FLAG tag, IRES element, and hrGFP were removed from the vector and replaced with a V5 tag, allowing for a C-terminal fusion to HDAC3 (plasmid MW91). To create the point mutation, we changed the nucleotides to code for a histidine residue in place of tyrosine at amino acid 298 (plasmid MW92). For the empty vector control, the HDAC3 coding sequence was not present, but all other elements remain (plasmid MW87). AAV was made by the Penn Vector Core and

the final titers were determined by qPCR (AAV-HDAC3(Y298H): $6.48 \times 10^{12}$ GC/ml; AAV-EV: $1.35 \times 10^{13}$ GC/ml).

**Lentivirus production.** For the CRISPR/dCas9 Synergistic Activation Mediator (SAM) system, lentiviral plasmids were purchased from Addgene for the dCas9-VP64-GFP (#61422-LVC) and MS2-P65-HSF1_Hygro (#61426-LVC) constructs (titers $\geq 8 \times 10^6$ TU/ml). For the *Per1* sgRNA, we cloned and inserted an antisense guide sequence corresponding to the CRE element in the *Per1* promoter (AGAGGGAGGTGACGTCAAAG) into the Addgene sgRNA(MS2) cloning backbone (#61427). The control sgRNA was identical, except that no guide sequence was cloned into the plasmid. Lentiviruses for both the *Per1* sgRNA (titer: $6.8 \times 10^7$ IFU/ml) and control sgRNA ($3.5 \times 10^7$ IFU/ml) were produced by the USC School of Pharmacy Lentiviral Laboratory.

For the pLVX-V5Per1 overexpression construct, we amplified full-length wildtype *Per1* from the Addgene pCMV-Sport2-mPer1 plasmid (#16203) and cloned the product into a modified pLVX-EF1α-IRES-mCherry backbone (Takara, #631987). The IRES and mCherry elements were removed and were replaced with a V5 tag, allowing for an N-terminal fusion to PER1 (plasmid MW206). For the empty vector (EV) control, the PER1 coding sequence was not present but all other elements remained (Plasmid MW93). Lentiviruses for the pLVX-v5Per1 (titer: $1.3 \times 10^8$ IFU/ml) and pLVX-EV (titer: $1.5 \times 10^8$ IFU/ml) were produced by the USC School of Pharmacy Lentiviral Laboratory. All lentiviral constructs were expressed under the EF1α promoter.

**Cell Culture Verification of pLVX-v5Per1 and CRISPR-SAM.** To verify that pLVX-v5Per1 produces overexpression of *Per1* mRNA, mouse HT22 cells (Salk Institute, La Jolla, CA, #T09031) were transfected with pLVX-v5Per1 or pLVX-EV (Fig. 5a) using Lipofectaine LTX (Invitrogen). Similarly, to verify that the CRISPR-SAM system can effectively drive *Per1* transcription, HT22 cells were transfected with dCas9-VP64 (lenti MS2-P65_HSF1_Blast was a gift from Feng Zhang Addgene plasmid #61425), MS2-P65-HSF1 (lenti dCAS-VP64_Blast was a gift from Feng Zhang, Addgene plasmid #61426), and either *Per1* sgRNA (Per1 sgRNA) or the non-targeting control sgRNA (ctrl sgRNA) (lenti sgRNA (MS2)_zeo backbone was a gift from Feng Zhang, Addgene plasmid #61427) using Lipofectamine LTX (Invitrogen). Cells were harvested after 24 or 48 h, lysed, and mRNA was isolated as described above. qRT-PCR was performed as described above using the *Per1*, *Per2*, and *Hes7* primers and probe listed in Table S4.

**siRNA.** For the *Per1* knockdown experiment, Accell SMARTpool small interfering RNAs (siRNAs; Dharmacon, GE) targeting *Per1* were diluted to a final total concentration of 10 μM in ddH20 and infused into the DH (1.0 μl/side). Accell non-targeting pool siRNA was used as the control (total concentration, 10 μM) and was infused in the same manner. For siRNA experiments, mice were handled and habituated as described above and surgery was performed the day after the final day of habituation. Mice were given a full day of recovery after surgery and were trained the following day (~48 h after surgery) to ensure maximal target knockdown. To ensure knockdown, mice were sacrificed ~1 h after test and punches from the dorsal hippocampus were processed with western blots to ensure knockdown of PER1 protein.

**Object location and object recognition memory tasks.** For object location and object recognition memory tasks, mice were handled for 2 min/day for 4 day and then habituated to the context for 5 min/day for six consecutive days in the absence of objects. During training, mice were exposed to two identical objects (100 ml beakers, spice tins, or glass candle holders) and allowed to explore for 10 min. During the retention test (24 h later for long-term memory or 60 m later for short-term memory), mice were allowed to explore for 5 min. For object location memory, one of the two familiar objects was moved to a new location. For object recognition memory, the object locations remained constant but one of the objects was replaced with a new item. Habituation for object recognition memory began at least one week after the completion of OLM testing and a new context and unfamiliar objects were used[48]. Exploration was scored when the mouse head oriented toward the object and came within 1 cm or when the nose touched the object. Total exploration time was recorded ($t$) and preference for the novel item was expressed as a discrimination index ($DI = (t_{novel} - t_{familiar}) / (t_{novel} + t_{familiar}) \times 100\%$). For training sessions, the object designated to be moved at test was used as the novel object to allow the training and testing DI to be directly compared. Mice that explored both objects for less than 2 s during testing or 3 s during training were removed from further analysis. Mice that showed a preference for one object during training ($DI > \pm 20$) were also removed. Habituation sessions were analyzed (to determine distance traveled and speed) using ANY-maze behavioral analysis software (Stoelting Co). All habituation, training, testing, and scoring were performed by experimenters blinded to the experimental groups.

**Context fear conditioning.** For contextual fear conditioning, mice were first handled for 5 days. During acquisition, mice were exposed to the context for 2 min and 28 s followed by a 2 s (0.75 mA) shock, a protocol that typically produces robust long-term memory[12,20]. Mice remained in the context for an additional 30 s before being removed. Mice were sacrificed 60 m after training along with homecage controls that were handled but not trained. Freezing behavior was measured using Ethovision 11 software (Noldus)[12].

**Elevated plus maze.** The plus-maze was conducted by an experimenter blind to the experimental groups. One week after the completion of ORM, a subset of mice were tested on the plus maze. Two arms of the maze were open ($30 \times 5$ cm) and two arms were enclosed ($30 \times 5 \times 15$ cm), connected by a central platform ($5 \times 5$ cm). The maze was elevated 40 cm above the floor. Mice were tested for 5 min on the apparatus, which consisted of placing each mouse onto the central platform facing one of the open arms. Between subjects, the maze was cleaned with 70% ethanol. The percentage of time spent in the closed and open arms was scored using ANY-maze software.

**Circadian rhythm analysis.** Young (3-m.o) and aging (18-m.o.) HDAC3$^{+/+}$ and HDAC3$^{flox/flox}$ mice were bred and housed under a 12 h light/dark (LD) cycle. Two weeks after AAV-CaMKII-Cre infusion (described above), mice were transferred to an isolated 12 h LD entrainment room for 7 days. Mice were then transferred into a light-protected activity analysis room, where locomoter activity was analyzed using optical beam motion dectors (Philips Respironics). Activity was monitored during 2 weeks of LD cycle entrainment in the light-protected room before mice were switched to constant darkness (DD) for an additional 3 weeks. Activity monitoring continued throughout the DD phase to determine whether HDAC3 deletion in the DH affected endogenous circadian rhythms. Data were collected using Minimitter VitalView 5.0 and Clocklab software (Actimetrics) was used to determine the onset of free activity. Tau values were calculated by obtaining the slope of this onset and calculating the least-squares fit with Clocklab software[49,50].

**Immunohistochemistry.** After the completion of behavior, mice were euthanized by cervical dislocation and their brains were removed and flash-frozen in ice-cold isopentane. Twenty micrometer slices were collected throughout the dorsal hippocampus, thaw-mounted on slides, and stored at $-80$ °C. Slides were fixed with with 4% paraformaldehyde (10-min), permeabilized in 0.01% Triton X-100 in 0.1 M PBS (5-min), and blocked for 1 h with 8% normal goat serum (Jackson). Slides were incubated overnight (4 °C) in rabbit antibody to HDAC3 (1:250, Abcam, clone Y415, ab32369) or V5 (1:1000, Abcam, ab9116), or chicken antibody to GFP (1:250, Aves Labs, #GFP1010). The following day, slides were washed and incubated for 1 h at room temperature with goat anti-rabbit Alexa 488 (HDAC3 and V5; 1:1000, ThermoFisher, #A-11008) or goat anti-chicken Alexa 488 (GFP; 1:1000, ThermoFisher, #A-11039) in the dark. Slides were then washed with PBST and incubated for 50 m in NeuroTrace 530/615 (1:50; ThermoFisher, #N21482), a fluorescent nissl stain. To quench nonspecific autofluorescence[51], slices were then washed in PBS with 0.01% Triton, rinsed in water, and incubated for 10-min in 10 mM CuSO$_4$ in 50 mM ammonium acetate buffer. Slices were again rinsed in water, washed in PBS and coverslipped with VectaShield Antifade mounting medium (Vector Laboratories).

All images were acquired with an Olympus Scanner VS110 with a 20x apochromatic objective (numerical aperture 0.75) with VS110 scanner software. All treatment groups were represented on each slide and all images on a slide were captured with the same exposure time under nonsaturating conditions. Immunolabeling intensity was quantified with ImageJ by sampling the optical density of the cell layer in CA1 and subtracting a sample of background fluorescence in the same image. For all AAV experiments, animals that failed to express the virus in area CA1 of the dorsal hippocampus were excluded from analyses. Imaging and quantification was performed by experimenters blind to the experimental conditions.

**Western blot.** To verify PER1 knockdown, Per1 siRNA and control siRNA mice were sacrificed 2 h after testing and brains were flash-frozen. Brains were coronally sectioned and 1 mm DH punches were collected from 500 μm slices. Punches were homogenized in T-PER buffer (Thermo Fisher) with Halt protease and phosphatase inhibitor (Thermo Fisher) using a dounce homogenizer. Protein lysates were quantified using a modified Bradford assay (BioRad) and 50 μg total protein lysate was loaded into each lane of a 7.5% NuPAGE Bis-Tris gel (Thermo Fisher). Gels ran for 50 min at 200 volts and blots were transferred overnight at 15 V at 4 °C onto nitrocellulose membranes (Novexi, LC2001). The following day, membranes were incubated in blocking buffer for 1 h (5% nonfat milk in Tris-buffered saline with 0.01% Tween 20), washed (0.1% Tween 20 in TBS) and then incubated in primary antibody (1:500, rabbit anti-Per1, Thermo Fisher #PA1-524) in primary antibody buffer (3% BSA in TBS with 0.1% Tween) overnight at 4 °C. The membranes were then washed and incubated in HRP-conjugated mouse antibody to rabbit (1:10,000, Jackson Laboratories, light chain-specific, #211-032-171) for 1 h. Membranes were washed and developed using Pierce SuperSignal West Pico Chemilumenescent Substrate (Pierce, 34077). Multiple film exposures were used to verify linearity. Blots were washed and stripped for 10 m with Restore Western Blot Stripping Buffer (Thermo Fisher #21059), washed again, and then re-probed overnight (4 °C) with a rabbit antibody to GAPDH (1:1000, Santa Cruz Biotechnology, SC25778). Full-length

PER1 and GAPDH blots are shown in Supplementary Figure 5. Relative optical densities were calculated from scanned film using ImageJ (US National Institutes of Health) by an experimenter blind to the experimental conditions. All values were normalized to GAPDH expression levels.

**qRT-PCR.** Tissue was collected from DH punches (described above) and frozen at −80 °C until processing. RNA was isolated using an RNeasy Minikit (Qiagen, #74104) and cDNA was created using the Transcriptor First Strand cDNA Synthesis kit (Roche, 04379012001). Primers and probes were derived from the Roche Universal ProbeLibrary (Table S4) and were used for multiplexing in the Roche Light-Cycle 480 II machine (Roche). All values were normalized to *Gapdh*. Analyses and statistics were performed using the Roche proprietary algorhithms and REST 2009 software based on the Pfaffl method[52,53].

**RNA sequencing.** RNA was isolated from dorsal hippocampus punches as described above, using the RNeasy minikit (Qiagen, 74104). RNA quality was assessed by Bioanalyzer and samples with an RNA integrity number >9 were included in our analysis. cDNA libraries for each group were prepared according to the TruSeq RNA Sample Preparation Guide (Illumina). Two hundred and fifty nanograms of total RNA from each mouse was purified with poly-T oligo-attached magnetic beads and heat fragmented. The first and second strand cDNA were then synthesized and purified. After the ends were blunted, the 3′ end was adenylated to prevent concatenation of the template during adapter ligation. For each group, a unique adapter set was added to the ends of the cDNA and the libraries were amplified by PCR. The quality of the library was assessed by Bioanalyzer and quantified using qRT-PCR with a standard curve prepared from a commercial sequencing library (Illumina). Samples were mutliplexed, with each behavioral group represented in each flow cell of the sequencer. 10 nM of each library was pooled in four multiplex libraries and sequenced on an Illumina HiSeq 2500 instrument during a single-read 50 bp sequencing, run by the Genomic High-Throughput Facility at the University of California, Irvine. The resulting sequencing data for each library were post-processed to produce FastQ files. The data were then demultiplexed and filtered using Illumina CASAVA 1.8.2 software as well as in-house software. Poor-quality reads (failing Illumina's standard quality tests) and control reads successfully aligned to the PhiX control genome were removed from analyses. The quality of the remaining sequences was further assessed using PHRED quality scores produced in real time during the base-calling step of the sequencing run (Supplementary Fig. 3a).

**Alignment to the reference genome and transcriptome.** The reads from each experiment were separately aligned to the reference genome and corresponding transcriptome using short-read aligners ELAND v2e (Illumina) and Bowtie[24]. Reads uniquely aligned by both tools to known exons or splice junctions with no more than two mismatches on any 25 bp fragment of the read were included in the transcriptome. Reads uniquely aligned, but with more than two mismatches on any 25 bp fragment of the read were removed from analyses. Similarly, reads matching several locations in the reference genome were removed from analysis. The percentage of reads assigned to the reference genome and transcriptome using this protocol is reported for each group of replicates (Supplementary Fig. 3b). This resulted in an average of approximately 30 million reads per sample.

To verify the presence of the pLVX and CRISPR-SAM plasmids following lentiviral infusion, we ran RNA sequencing on hippocampal samples from a subset of three animals per group following behavior. As the number of cells infected with the lentiviruses in vivo was too small to reliably detect the presence of the appropriate transcripts using RT-qPCR (Supplementary Fig. 6a, d), we used RNA-seq as a more sensitive method to detect the presence of these transcripts. An updated version of genome assembly and genome annotation were built by appending the sequences and annotation of the plasmid constructs to the original mm10 mouse genome and to the mm10 mouse genome annotation, respectively. Using the reads alignment tool TopHat in the the Tuxedo Suite[54], reads were mapped to the genome and only hits that were unique to the plasmid transcripts in comparison with the endogenous mouse reference genome were considered "matched reads" to the plasmid constructs. The number of those matched reads were then quantified for each construct for each sample.

**Gene expression and differential analysis.** Gene expression levels were directly computed from the read alignment results for each replicate. Standard RPKM values[55], reads per kilobase of exon model per million mapped reads) were extracted for each gene covered by the sequencing data and each replicate used in this study.

Differential transcriptional analyses were performed using Cyber T[56,57] across each pair of groups (homecage versus 60 m after training) to identify genes up- or down-regulated after OLM. In addition to the 18-month-old HDAC3[+/+] and HDAC3[flox/flox] mice trained for the current study, identically processed RNA-sequencing data from 3-month old C57 wildtype mice (homecage and OLM-trained in the same manner as the current study) from a

previous study[20] was used for differential analyses. The number of animals (biological replicates) for each group was: Young HDAC3[+/+] HC: 6, Young HDAC3[+/+] OLM: 6, Old HDAC3[+/+] HC: 6, Old HDAC3[+/+] OLM: 6, Old HDAC3[flox/flox] HC: 8, Old HDAC3[flox/flox] OLM: 8. No technical replicates were used. The p-value threshold for determining differential expression is 0.05 for all groups. False Discovery Rate (FDR) as measured by the Benjamini & Hochberg (BH) *q*-value was used to correct for multiple testing, with a FDR threshold of 0.15. The sets of genes upregulated after experience (compared to homecage) for these 3 groups (young wildtype, aging HDAC[+/+], or aging HDAC3[flox/flox]) were intersected to determine genes common to two or more groups. Enrichment of each group for tissue-specific expression, Gene Ontology terms[58], and KEGG pathways[59,60] was assessed using DAVID[61], based on differentially expressed genes after behavior. Data visualization was performed using "matplotlib" for python and "ggplot" for R.

**Chromatin immunoprecipitation.** ChIP was performed on DH punches based on the protocol from the Millipore ChIP kit[11,12,62]. Tissue was cross-linked with 1% formaldehyde (Sigma), lysed and sonicated, and chromatin was immunoprecipitated overnight with 2 µl of anti-H4K8Ac (Millipore #17-10099) or 4 µl of anti-HDAC3 (Millipore #17-10238) or an equivalent amount of Normal Rabbit Serum (H4K8Ac negative control, Millipore) or anti-mouse IgG (HDAC3 negative control, Millipore). After washing, chromatin was eluted from the beads and reverse cross-linked in the presence of proteinase K before column purification of DNA. Primer sequences for the promoters of each gene were designed by the Primer 3 program (Table S4). Five µl of input, anti-H4K8Ac IgG/anti-HDAC3 IgG, or anti-rabbit/mouse IgG immunoprecipitate from each animal were examined in duplicate. To normalize ChIP-qPCR data, we used the percent input method. The input sample was adjusted to 100% and both the IP and IgG samples were calculated as a percent of this input using the formula $100*AE^{(adjusted\ input\ –\ Ct(IP))}$. Fold enrichment was then calculated as a ratio of the ChIP to the average IgG. An in-plate standard curve determined amplification efficiency (AE).

**Slice preparation and recording.** Young (approximately 3-m.o.) and aging (18-m.o.) mice were stereotaxically infused with virus. For the first experiment, young and old HDAC3[+/+] and HDAC3[flox/flox] mice were infused with AAV-CaMKII-Cre bilaterally into the DH. For the second experiment, young and old wildtype C57 mice were infused with AAV-HDAC3(Y298H) into the DH of one hemisphere and AAV-EV control into the other hemisphere. Two weeks after infusion (to allow for optimal virus expression)[2], transverse hippocampal slices (320 µm) were placed in an interface recording chamber with preheated (31 ± 1 °C) artificial cerebrospinal fluid (124 mM NaCl, 3 mM KCl, 1.25 mM KH₂PO₄, 1.5 mM MgSO₄, 2.5 mM CaCl₂, 26 mM NaHCO₃, and 10 mM D-glucose). Slices were continuously perfused at a rate of 1.75–2 ml per min while the surface of the slices was exposed to warm, humidified 95% O₂/5% CO₂. Recordings began after at least 2 h of incubation.

Field excitatory postsynaptic potentials (fEPSPs) were recorded from CA1b stratum radiatum using a single glass pipette (2–3 MΩ) filled with 2 M NaCl. Stimulation pulses (0.05 Hz) were delivered to Schaffer collateral-commissural projections using a bipolar stimulating electrode (twisted nichrome wire, 65 µm) positioned in CA1c. Current intensity was adjusted to obtain 50% of maximal fEPSP response. After a stable baseline was established, LTP was induced by a single train of five theta bursts, in which each burst (four pulses at 100 Hz) was delivered 200 ms apart (i.e., at theta frequency). The stimulation intensity was not increased during TBS. Data were collected and digitized by NAC 2.0 Neurodata Acquisition System (Theta Burst).

**Statistical analysis.** Statistical analyses were conducted as indicated in the text and figure legends using Prism 6 (GraphPad). Our analytic approaches are based on previously published work[12,20,63,64]. No statistical methods were used to pre-determine sample sizes, but our sample sizes are similar to those generally used in the field, including those reported in previous publications[12,20,64,65]. Data distribution was assumed to be normal, with similar variance observed among groups, but this was not formally tested. When an experiment had two groups to compare, two-tailed Student's *t*-tests were used. When two factors where compared, (such as age and genotype or session and group), data were analyzed with two-way ANOVAs followed by Sidak's multiple comparisons post hoc tests. All analyses are two-tailed, with an *α* value of 0.05 required for significance. Error bars in all figures represent SEM. For all experiments, values ± 2SD from the group mean were considered outliers and were removed from analyses.

**Data availabilty.** The data supporting the findings of this study are available from the corresponding author upon reasonable request. RNA sequencing data have been deposited in NCBI's Gene Expression Omnibus and are accessible through GEO Series accession number GSE94832.

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

## Acknowledgements

We wish to thank all members of the Wood lab for scientific discussions and technical assistance. We would also like to acknowledge the University of California, Irvine Institute for Genomics and Bioinformatics High-Throughput Facility for help with RNA sequencing and Yuzo Kanomata for additional computing support. This work was supported by the National Institutes of Health grants (MH101491, AG051807, and AG050787 to M.A.W. and T32-AG000096-31, F32-AG052303, and K99-AG056596 to J.L.K.). The work of Y.L., C.M., S.C., and P.B. was in part supported by NSF grant IIS- 1550705, DARPA grant D17AP00002, and NIH GM123558 to P.B. E. M. was supported by a long-term EMBO post-doctoral fellowship and an ARC Foundation award and work in the P.S.-C. lab was supported by grants from the National Institutes of Health, INSERM, and the Novo Nordisk Foundation Challenge Program.

## Author contributions

J.L.K., D.P.M. and M.A.W. designed the experiments. J.L.K., Y.A., A.V.C. and D.P.M. conducted the experiments. J.L.K. and M.A.W. wrote the manuscript. E.A.K. conducted the electrophysiology and analyzed the results. J.L.K., E.M. and P.S.-C. designed and conducted the circadian rhythm experiment. Y.L., C.N.M., S.C. and P.B. performed the RNA sequencing analysis. A.J.L., A.O.W., G.S., D.R. and C.M.M. contributed to experiments. P.B., P.S.-C. and D.P.M. assisted in experimental design, data analysis, and manuscript preparation.

## Additional information

**Competing interests:** The authors declare no competing interests.

