## [Peer Review File · Nature Communications]

Reviewers' comments:

Reviewer #1 (Remarks to the Author):

This nice story traces the role of hippocampal HDAC3 expression in long-term spatial memory in ageing mice, and ultimately attributes a novel and significant role to the regulation of the clock gene *Per1* by HDAC3. The idea that HDAC3 influences long-term memory, including during ageing, is not new. The idea that clock genes affect memory is also not new. However, the idea that HDAC3 could act via expression of a clock gene, independent of overall circadian rhythmicity, is definitely new. The data is of high technical quality, and the results are convincing. Therefore, I strongly support its publication, and I do not think that further experiments are needed. However, in several cases the presentation is misleading, overstated, or both, and I think that this should be corrected before publication.

p. 5 – “To determine whether HDAC3 limits memory formation in the aging brain”
If deleting HDAC3 in young mice can restore memory, it is highly likely that doing the same in old mice will, too (unless HDAC3 happens not to be expressed in old mice). This experiment is basically a necessary control for what comes after, but trying to make it sound more important mars the impact of the other nice results. I would tone this down.

p. 11 – “To rule this out, we assessed the circadian rhythmicity of young (3-m.o.) and aging (18-m.o.) HDAC3^{flox/flox} mice and their HDAC3^{+/+} littermates following AAV-CaMKII-Cre infusion.”

It is clear that circadian rhythmicity of behavior is not changed, but the authors did not address circadian rhythmicity in the hippocampus. I do not suggest that they go back and do this (unless they happen to have the data already), but this should be stated as an important caveat to their results here and in their discussion. It is perfectly possible that circadian rhythmicity in a given tissue is altered without altering global rhythmicity of behavior.

p. 11 -- “To test whether hippocampal *Per1* is required for long-term memory formation, we assessed whether hippocampus-dependent learning typically induces *Per1* mRNA expression.”

This is establishing only a correlation, not a requirement. Later, they establish a likely necessity for *Per1* by re-expressing it and documenting a memory improvement.

I think that the discussion would be improved by adding in a discussion of the authors' conclusions about the role of *Per1* in comparison with others' results implying that the effects of HDAC3 upon memory might be mediated by NF κ B or FMRP (Sharma 2015, cited by the authors, and Franklin 2014). I think that the authors might be able to make a convincing case that *Per1* is upstream of both of these.

Secondly, it is possible that HDAC3 is acting via *Per1*, which is why readers will find this paper interesting. Restoring *Per1* expression and improving memory supports the authors' conclusions. However, the transcriptomic changes are complex, and other untested genes

also play known roles in memory. Therefore, it is possible that the results are a bit more complex than portrayed here. I think that this could and should be acknowledged in the discussion, and it would not take away from the authors' contributions.

Minor problems

I would recommend softening the title: "Mediates" implies to me that it is the only factor, which is not rigorously shown here. "Contributes to" suggests that it plays a role, which the authors do elegantly show.

p.3 "suggesting that these processes might share similar mechanisms": I think that the authors probably mean regulation by common components, instead of actually sharing mechanisms.

Reviewer #2 (Remarks to the Author):

Although it is well understood that both circadian rhythms and biological aging affect memory function, few studies have attempted to identify links between these two core processes at a molecular level. This manuscript by Kwapis, et al examines the potential role of HDAC3, a histone deacetylase, and Per1, a gene implicated in circadian rhythms, in hippocampus-dependent learning and memory, long-term potentiation, and experience-dependent changes in gene expression. Overall, the manuscript provides a convincing demonstration that HDAC3 (and regulation of Per1 by HDAC3) is critical for multiple aspects of hippocampal function, including long-term potentiation and hippocampus-dependent memory formation. Major strengths of the manuscript are the application of multiple orthogonal approaches (e.g., conditional deletion or pharmacological inhibition of HDAC3 in separate experiments), examination of HDAC3 and Per1 regulation in both young and old animals, and application of RNA-seq to understand transcriptional consequences of HDAC3 conditional deletion. Weaknesses of the manuscript, although somewhat minor, include the lack of proper validation and control experiments for some figures, lack of essential information on experimental design, and occasional over-interpretation of available data. These concerns are outlined below.

Major comments:

1. One of the repeating arguments in the introduction and discussion of this manuscript is the concept that nuclear genomes in neurons in the aged hippocampus exist in a transcriptionally repressed state, which prevents the induction of genes necessary for memory formation. Although the observation that HDAC3 deletion or disruption rescues memory and LTP is taken to be consistent with this hypothesis, several results presented in this manuscript are at odds with this interpretation. For example, older animals actually exhibited more (not less) up and down-regulated genes in response to OLM training (Fig. 3C). However, even if this hypothesis had been supported by RNA-seq results, there is little evidence to tie HDAC3 to this repressive regulation during aging – HDAC3 levels do not seem to increase in aged animals, and HDAC3 deletion does not rescue the vast majority of

genes that are induced by OLM training in young animals but not old animals. Thus, the “repressive chromatin” hypothesis seems to be somewhat forced on the data, which distracts some from the more exciting findings. This general argument should be changed or potentially even removed.

2. For all object tasks, statistical verification should be provided to support claims of intact or non-intact memory in addition to the group differences reported by the authors. For example, if OLM memory is reported as intact for a specific group, the authors should include statistics demonstrating a significant preference for the object in the novel location in addition to between group differences that are already reported. For claims that memory is not intact, the authors should include stats demonstrating there is no significant difference between time spent with the moved and unmoved objects. While the author’s claims generally seem to be supported in the data (discrimination index scores are close to zero for “impaired” groups or further from zero for “intact” groups), this could be supported in some way.

3. Use of the term “learning-induced” changes in gene expression throughout the manuscript is not really appropriate with the control groups currently used. For the authors to make claims of learning-induced changes, this would at the very least require a context only control that has received the same exposure to the behavioral arena but without exposure to objects during the training session. Tempering the language by using something more appropriate given the current use of homecage controls (perhaps “experience-induced”) would be more accurate.

4. The CRISPR/dCas9 targeting experiments in Figure 5c are innovative, but seem preliminary for this manuscript. CRISPR-induced increases in *Per1* are not verified in any neuronal system, and it is not clear if lentiviral constructs are expressed after CA1 infusion. Overall, while this approach is a nice addition, there are several controls and validation experiments that must be performed here. Proper controls might include a non-targeting gRNA (instead of just an empty vector). Additionally, it is important to determine whether this effect is specific to *Per1*, or whether other genes linked to learning and memory are impacted as well. Since the manuscript already includes a more conventional overexpression experiment, I would suggest either adding these controls or removing the CRISPR results entirely.

5. The manuscript methods section is missing key information about the number of distinct samples used for RNA-seq, the number of reads obtained per sample (or average number of reads), the methodology used for multiple-comparisons correction, and the number of biological and technical replicates that contributed to statistical analysis here. All of this information should be included.

6. The authors frequently refer to the concept that circadian rhythms affect long-term memory, but suggest that this manuscript reveals an autonomous role for clock genes in memory outside of the function of the central circadian clock. However, the authors have not fully dissected the idea that learning-induced changes in *Per1* could alter local clock gene patterns in the CA1, or that CA1 changes in *Per1* are fully autonomous of SCN-generated rhythms. While Supplementary Figure 4 shows that HDAC3 deletion in the CA1 does not alter circadian rhythms in young or old animals, this control is not done for any manipulations of *Per1*. Similarly, although control behavioral data (object recognition memory, distance travelled, exploration, and anxiety-like behavior) is provided for HDAC3 manipulations, there is no similar control data provided for *Per1* manipulations. If this is not

readily available, the authors should soften their conclusions to reflect this weakness and add discussion of the potential caveat that CA1 Per1 manipulations could have affected circadian rhythms.

Minor comments:

1. At least two previous manuscripts – Halder, et al 2015 Nature Neuroscience (PMID 26656643) and Duke, et al 2017 Learning and Memory (PMID 28620075) have reported upregulation of Per1 mRNA in the CA1 of the hippocampus during contextual fear learning. These publications demonstrated neuron-specific increases in Per1 (Halder, et al), and extend this basic finding to another species (rats were used in Duke, et al). These findings support the line of investigation pursued in this manuscript, and should be mentioned.
2. Although the authors state the sex ratio for each experiment in the figure legend, there are no formal comparisons of male and female datapoints. In the absence of this, it would be useful to represent male and female datapoints using different symbols so that readers can evaluate potential sex differences.
3. Figure 4C: Did a decrease of HDAC3 occupancy at Per1 after training correlate with an increase in acetylation or Per1 expression shown in Figure. 3G-H?
4. Supplementary Figure 1 B: No explanation of EV and V5 on the figure or the figure legend. The IHC only illustrates the expression of V5 virus in the hippocampus, it does not actually prove that the catalytic activity of HDAC3 has been perturbed. This may have been done in previous studies that were cited. Supplementary Figure 1 D: No explanation of blue bars on the figure.
5. The authors should clarify if they have evaluated exploration during habituation and anxiety for both the HDAC3flox/flox and the HDAC3(Y298H) groups, and if so, include data for both of these groups in Supplementary Fig. 2, rather than only for the HDAC3flox/flox group.
6. On page 10: “Together these results suggest that deleting HDAC3 restores acetylation at the Per1 promoter and expression of Per1 mRNA in response to learning.” “Restore” suggests levels of H4K8 acetylation are being returned to a previously observed state, or perhaps to levels found after training in young WT controls, which there is not evidence to support here. The sentence should be rephrased to more accurately describe the data being reported.

Response to Reviews

We sincerely thank the reviewers for the time and effort they dedicated to providing us with their insightful comments. We have addressed the reviewers' concerns below and have revised the manuscript to incorporate their suggestions for improvement. Relevant changes in the manuscript text are shown in red font. The manner in which each specific criticism was addressed is explained in detail below. Reviewer comments are in bolded italics.

Itemized list of major changes:

- OLM test sessions graphs now include training DIs to support claims of intact or non-intact memory by comparing test DI to training DI.
 - Statistics for this session have been updated to allow this comparison
- Graphs from all experiments including females have been updated with different symbols for males (black circles) and females (gray squares).
- Both *Per1* overexpression experiments (Fig. 5) were replicated and the data are included.
- CRISPR experiment has additional data validating the system, including measurement of *Per2* and *Hes7* mRNA, immunofluorescence to verify expression *in vivo*, and RNA-seq to detect plasmids in the hippocampus after behavior.
- pLVX-*Per1* experiment has additional data validating the system, including measurement of *Per2* and *Hes7* mRNA, immunofluorescence to verify expression *in vivo*, and RNA-seq to detect plasmids in the hippocampus after behavior.
- All ChIP data has been replicated and the data are included
- Added discussion about whether changes in *Per1* might alter molecular oscillations within the dorsal hippocampus
- Changed title, replacing “Mediates” with “Contributes to.”
- Added a discussion of NF κ B, FMRP, and other untested genes identified here that may underlie HDAC3's effects on memory in addition to *Per1*.
- Removed argument that HDAC3 contributes to a repressive chromatin structure in the aging brain that limits gene expression.
- Included more methodological details for RNA-seq study

Reviewer 1:

1. *p. 5 – “To determine whether HDAC3 limits memory formation in the aging brain” If deleting HDAC3 in young mice can restore memory, it is highly likely that doing the same in old mice will, too (unless HDAC3 happens not to be expressed in old mice). This experiment is basically a necessary control for what comes after, but trying to make it sound more important mars the impact of the other nice results. I would tone this down.*

We have reworded this sentence (p.5): “To determine whether deletion of HDAC3 improves memory in aging mice.”

2. *p. 11 – “To rule this out, we assessed the circadian rhythmicity of young (3-m.o.) and aging (18-m.o.) HDAC3^{lox/lox} mice and their HDAC3^{+/+} littermates following AAV-CaMKII-Cre infusion.”*

It is clear that circadian rhythmicity of behavior is not changed, but the authors did not address circadian rhythmicity in the hippocampus. I do not suggest that they go back and do this (unless they happen to have the data already), but this should be stated as an important caveat to their results here and in their discussion. It is perfectly possible that circadian rhythmicity in a given tissue is altered without altering global rhythmicity of behavior.

We agree that it is possible that our hippocampal manipulations may have affected molecular oscillations of other clock genes and plasticity-related molecules may within the hippocampus itself. While we do not have the data to include this in the current study, we do plan to address these questions in future work. We now discuss this possibility and our plan to investigate this in future work on pp. 19-20.

- 3. p.11 -- "To test whether hippocampal Per1 is required for long-term memory formation, we assessed whether hippocampus-dependent learning typically induces Per1 mRNA expression."
This is establishing only a correlation, not a requirement. Later, they establish a likely necessity for Per1 by re-expressing it and documenting a memory improvement.***

We initially meant that the next series of studies (not just the next experiment) were testing whether hippocampal *Per1* is required for long-term memory. We now reword this part (p. 11): "We next wanted to determine whether hippocampal *Per1* is required for long-term memory formation. First, we assessed whether hippocampus-dependent learning typically induces *Per1* mRNA expression."

- 4. I think that the discussion would be improved by adding in a discussion of the authors' conclusions about the role of Per1 in comparison with others' results implying that the effects of HDAC3 upon memory might be mediated by NFkB or FMRP (Sharma 2015, cited by the authors, and Franklin 2014). I think that the authors might be able to make a convincing case that Per1 is upstream of both of these.***

Secondly, it is possible that HDAC3 is acting via Per1, which is why readers will find this paper interesting. Restoring Per1 expression and improving memory supports the authors' conclusions. However, the transcriptomic changes are complex, and other untested genes also play known roles in memory. Therefore, it is possible that the results are a bit more complex than portrayed here. I think that this could and should be acknowledged in the discussion, and it would not take away from the authors' contributions.

We thank the reviewer for these thoughtful insights! To address the possible interactions between HDAC3-mediated regulation of *Per1* and other mechanisms downstream of HDAC3 like FMRP and NF- κ B, we have added a paragraph to the discussion (p. 17). This discussion covers the possible role of PER1 in regulating CREB phosphorylation and the likelihood that other genes (including the untested genes identified through our RNA-seq) are contributing to the memory-enhancing effects of HDAC3 deletion.

5. *I would recommend softening the title: “Mediates” implies to me that it is the only factor, which is not rigorously shown here. “Contributes to” suggests that it plays a role, which the authors do elegantly show.*

Thank you for the suggestion. We have changed the title to incorporate this wording.

6. *p. 3 “suggesting that these processes might share similar mechanisms”: I think that the authors probably mean regulation by common components, instead of actually sharing mechanisms.*

We have changed the wording here to, “suggesting that common molecular mechanisms might underlie both processes.”

Reviewer 2:

1. *One of the repeating arguments in the introduction and discussion of this manuscript is the concept that nuclear genomes in neurons in the aged hippocampus exist in a transcriptionally repressed state, which prevents the induction of genes necessary for memory formation. Although the observation that HDAC3 deletion or disruption rescues memory and LTP is taken to be consistent with this hypothesis, several results presented in this manuscript are at odds with this interpretation. [...] Thus, the “repressive chromatin” hypothesis seems to be somewhat forced on the data, which distracts some from the more exciting findings. This general argument should be changed or potentially even removed.*

We appreciate this point and agree with the reviewer that the argument is not a good fit for our data. We have removed this argument except when presenting our initial hypothesis, as we began this series of experiments with the hypothesis that HDAC3-mediated repression of gene expression would underlie age-related impairments in long-term memory formation. On page 8, for example, before we ran RNA-seq, we expected to see HDAC3-mediated reductions in gene expression in the aging brain. We have removed this statement elsewhere, including from the discussion, as we agree that our data do not entirely fit this conclusion.

2. *For all object tasks, statistical verification should be provided to support claims of intact or non-intact memory in addition to the group differences reported by the authors. For example, if OLM memory is reported as intact for a specific group, the authors should include statistics demonstrating a significant preference for the object in the novel location in addition to between group differences that are already reported. For claims that memory is not intact, the authors should include stats demonstrating there is no significant difference between time spent with the moved and unmoved objects. While the author’s claims generally seem to be supported in the data (discrimination index scores are close to zero for “impaired” groups or further from zero for “intact” groups), this could be supported in some way.*

To address this point, we have re-graphed our behavioral data to show the discrimination index for both training and testing on the same graph. This allows the reader to see whether a group learned, relative to their baseline preference (DI) for the objects at training. We have also included the appropriate statistics (two-way ANOVA followed by Sidak's *post hoc* tests) to compare the training and testing DIs for each group to determine whether there was significantly more preference at test relative to training (suggesting intact memory) or whether preference at training and testing was similar (suggesting non-intact memory).

- 3. Use of the term “learning-induced” changes in gene expression throughout the manuscript is not really appropriate with the control groups currently used. For the authors to make claims of learning-induced changes, this would at the very least require a context only control that has received the same exposure to the behavioral arena but without exposure to objects during the training session. Tempering the language by using something more appropriate given the current use of homecage controls (perhaps “experience-induced”) would be more accurate.**

We appreciate this point and have changed “learning-induced” and similar phrases to “experience-induced” throughout the manuscript for clarity.

- 4. The CRISPR/dCas9 targeting experiments in Figure 5c are innovative, but seem preliminary for this manuscript. CRISPR-induced increases in *Per1* are not verified in any neuronal system, and it is not clear if lentiviral constructs are expressed after CA1 infusion. Overall, while this approach is a nice addition, there are several controls and validation experiments that must be performed here. Proper controls might include a non-targeting gRNA (instead of just an empty vector). Additionally, it is important to determine whether this effect is specific to *Per1*, or whether other genes linked to learning and memory are impacted as well. Since the manuscript already includes a more conventional overexpression experiment, I would suggest either adding these controls or removing the CRISPR results entirely.**

We think that the manuscript is stronger with two complementary methods of *Per1* overexpression and therefore have run some additional experiments to improve our validation of the CRISPR-SAM system used to drive *Per1*. **First**, we demonstrated in hippocampal cell culture (HT22 cells) that CRISPR-SAM transfection drives a significant increase in *Per1* mRNA (Fig. 5F). **Next**, to verify that the CRISPR system is expressing *in vivo* in our behavioral experiment, we ran immunofluorescence to detect the GFP tag on the dCas9-VP64-GFP construct (Supplementary Figure 6D). The spread of the lentiviruses was restricted to a very small region of the dorsal hippocampus (area CA1b, known to be critically important for long-term memory for OLM (see Barrett et al., 2011, Fig. 1a)). The viral spread was too small to detect the constructs using RT-qPCR, so we used RNA sequencing to verify the presence of the constructs (both the CRISPR and pLVX constructs) in a small subset (n=3/group) of samples (Supplementary Fig. 7). **Finally**, to determine the specificity of the CRISPR-SAM system, we also verified that there was no change in expression of *Per2* (another Period family member, Supplementary Fig. 6E) or *Hes7* (the nearest downstream gene, Supplementary Fig. 6F).

We also ran these experiments for the pLVX-Per1 system and notably found that *Per2* mRNA was also significantly increased by pLVX-Per1 (Supplementary Fig. 6B), although this increase was far smaller than the observed increase in *Per1* (Fig. 5B). As the CRISPR system had both a more physiologically relevant increase in *Per1* (more similar to the increases in *Per1* observed after behavior, e.g. Fig. 4A, 4B) and did not drive this off-target increase in *Per2*, we believe that including this manipulation in the paper adds important information to our conclusions about the role of *Per1* in memory. We discuss this pLVX-Per1-mediated increase in *Per2* on pp. 17-18.

Finally, we want to apologize for being unclear in our previous draft of this manuscript. The control for the CRISPR experiments was initially called an “empty vector,” but this was a bit misleading. The control sgRNA is identical to the *Per1* sgRNA, except that it lacks a 20 nucleotide sequence between the MS2 loops that targets the sgRNA to *Per1*. To more accurately reflect this, we now refer to this as the “control sgRNA” rather than “empty vector.” We put in a sentence to clarify this on p. 14 and in the Methods.

- 5. The manuscript methods section is missing key information about the number of distinct samples used for RNA-seq, the number of reads obtained per sample (or average number of reads), the methodology used for multiple-comparisons correction, and the number of biological and technical replicates that contributed to statistical analysis here. All of this information should be included.***

We apologize for this oversight and now include this information in the Methods section (p. 31) and in the caption for Fig. 3 (p. 46). We consider each animal to be an n of 1, making the group sizes: Young HDAC^{+/+} HC: n=6(3F), Young HDAC3^{+/+} OLM: n=6(3F), Old HDAC3^{+/+} HC: n=6(2F), Old HDAC3^{+/+} OLM: n=6(2F), Old HDAC3^{flx/flx} HC: n=8(5F), Old HDAC3^{flx/flx} OLM: n=8(5F).

- 6. The authors frequently refer to the concept that circadian rhythms affect long-term memory, but suggest that this manuscript reveals an autonomous role for clock genes in memory outside of the function of the central circadian clock. However, the authors have not fully dissected the idea that learning-induced changes in Per1 could alter local clock gene patterns in the CA1, or that CA1 changes in Per1 are fully autonomous of SCN-generated rhythms. While Supplementary Figure 4 shows that HDAC3 deletion in the CA1 does not alter circadian rhythms in young or old animals, this control is not done for any manipulations of Per1. Similarly, although control behavioral data (object recognition memory, distance travelled, exploration, and anxiety-like behavior) is provided for HDAC3 manipulations, there is no similar control data provided for Per1 manipulations. If this is not readily available, the authors should soften their conclusions to reflect this weakness and add discussion of the potential caveat that CA1 Per1 manipulations could have affected circadian rhythms.***

We agree that it is possible that learning-induced changes in *Per1* or our virus-mediated overexpression of *Per1* may altered local clock gene patterns within the dorsal

hippocampus. To address this, as well as the possibility that our *Per1* manipulations may have affected the core circadian clock, we have added a paragraph to the discussion (pp. 19-20) that addresses this possibility. We do plan to address these questions in future work, as mentioned in the discussion.

Additionally, to address the reviewer's other concern, we have now provided habituation data for each behavioral experiment (Supplementary Fig. 8) to show that activity (distance traveled) is not affected by our manipulations. Exploration time during the test session is also shown for each behavioral experiment. Unfortunately, we did not run the elevated plus maze task for each experiment and thus cannot show that information here.

- 7. At least two previous manuscripts – Halder, et al 2015 Nature Neuroscience (PMID 26656643) and Duke, et al 2017 Learning and Memory (PMID 28620075) have reported upregulation of Per1 mRNA in the CA1 of the hippocampus during contextual fear learning. These publications demonstrated neuron-specific increases in Per1 (Halder, et al), and extend this basic finding to another species (rats were used in Duke, et al). These findings support the line of investigation pursued in this manuscript, and should be mentioned.***

Thank you for bringing these studies to our attention. We now discuss them on p. 19.

- 8. Although the authors state the sex ratio for each experiment in the figure legend, there are no formal comparisons of male and female datapoints. In the absence of this, it would be useful to represent male and female datapoints using different symbols so that readers can evaluate potential sex differences.***

We now use different symbols to represent males and females in all graphs, with males represented by black circles and females represented by gray squares. We appreciate this clever suggestion!

- 9. Figure 4C: Did a decrease of HDAC3 occupancy at Per1 after training correlate with an increase in acetylation or Per1 expression shown in Figure. 3G-H?***

We tested this but did not find any significant correlations here and thus did not include these statistics in the manuscript.

- 10. Supplementary Figure 1 B: No explanation of EV and V5 on the figure or the figure legend. The IHC only illustrates the expression of V5 virus in the hippocampus, it does not actually prove that the catalytic activity of HDAC3 has been perturbed. This may have been done in previous studies that were cited. Supplementary Figure 1 D: No explanation of blue bars on the figure.***

We apologize for the lack of clarity in this figure. These issues have all been fixed on the figure. We have validated HDAC3(Y298H) as blocking the catalytic activity of HDAC3 in a previous publication (Kwapis et al., 2017, referenced in the text (top of p. 5)).

- 11. *The authors should clarify if they have evaluated exploration during habituation and anxiety for both the HDAC3flox/flox and the HDAC3(Y298H) groups, and if so, include data for both of these groups in Supplementary Fig. 2, rather than only for the HDAC3flox/flox group.***

We now include habituation graphs for every behavioral experiment (Supplementary Fig. 8), including habituation for the HDAC3(Y298H) group. We did not evaluate anxiety for the HDAC3(Y298H) animals. As the complete deletion of HDAC3 did not affect anxiety-like behavior, we think it is unlikely that the more precise disruption of HDAC3 activity with HDAC3(Y298H) would affect anxiety levels.

- 12. *On page 10: “Together these results suggest that deleting HDAC3 restores acetylation at the Per1 promoter and expression of Per1 mRNA in response to learning.” “Restore” suggests levels of H4K8 acetylation are being returned to a previously observed state, or perhaps to levels found after training in young WT controls, which there is not evidence to support here. The sentence should be rephrased to more accurately describe the data being reported.***

We have changed this sentence to read, “Together, these results suggest that deleting HDAC3 increases acetylation at the *Per1* promoter and expression of *Per1* mRNA in response to learning” to more accurately reflect our results.

REVIEWERS' COMMENTS:

Reviewer #1 (Remarks to the Author):

I apologise to the authors for my delay in re-reviewing their manuscript, but I had stated that I did not need to see the manuscript again anyhow. My comments requested only a variety of qualifying statements from the authors about their interesting results. They have implemented all of these, and thereby satisfied all of my concerns. In rereading the manuscript, I also think that they have addressed the minor concerns of other reviewers adequately.

Reviewer #2 (Remarks to the Author):

The authors have succeeded in addressing all of my previous concerns. This is a well written manuscript that was a pleasure to read and should have a significant impact on the field.